# Inhibitory role of Annexin A1 in pathological bone resorption and therapeutic implications in periprosthetic osteolysis

Hend Alhasan[1], Mohamad Alaa Terkawi [1✉], Gen Matsumae[1], Taku Ebata[1], Yuan Tian[1],
Tomohiro Shimizu [1✉], Yoshio Nishida[1], Shunichi Yokota[1], Fayna Garcia-Martin[2,3], Mahmoud M. Abd Elwakil[4],
Daisuke Takahashi[1], Mahmoud A. Younis [4], Hideyoshi Harashima[4], Ken Kadoya[1] & Norimasa Iwasaki[1]

There is currently no therapy available for periprosthetic osteolysis, the most common cause of arthroplasty failure. Here, the role of AnxA1 in periprosthetic osteolysis and potential therapeutics were investigated. Reducing the expression of AnxA1 in calvarial tissue was found to be associated with increased osteolytic lesions and the osteolytic lesions induced by debris implantation were more severe in AnxA1-defecient mice than in wild-type mice. AnxA1 inhibits the differentiation of osteoclasts through suppressing NFκB signaling and promoting the PPAR-γ pathway. Administration of N-terminal-AnxA1 (Ac2-26 peptide) onto calvariae significantly reduced osteolytic lesions triggered by wear debris. These therapeutic effects were abrogated in mice that had received the PPAR-γ antagonist, suggesting that the AnxA1/PPAR-γ axis has an inhibitory role in osteolysis. The administration of Ac2–26 suppressed osteolysis induced by TNF-α and RANKL injections in mice. These findings indicate that AnxA1 is a potential therapeutic agent for the treatment of periprosthetic osteolysis.

---

[1] Department of Orthopedic Surgery, Faculty of Medicine and Graduate School of Medicine, Hokkaido University, Kita-15, Nish-7, Kita-ku, Sapporo 060-8638, Japan. [2] Graduate School and Faculty of Advanced Life Science, Laboratory of Advanced Chemical Biology, Hokkaido University, N21 W11, Kita-21, Nish-11, Kita-ku, Sapporo 001-0021, Japan. [3] Faculty of Science and Technology, Department of Chemistry, University of La Rioja, E-26006 Logroño, Spain. [4] Laboratory of Innovative Nanomedicine, Faculty of Pharmacy and Pharmaceutical Sciences, Hokkaido University, Kita-12 Nishi-6, Kita-Ku, Sapporo 060-0812, Japan. ✉email: materkawi@med.hokudai.ac.jp; simitom@wg8.so-net.ne.jp

Total joint arthroplasty (TJA) is the most reasonable approach for orthopedic surgeons, since it reduces pain and restores the function of end-stage arthritic joints. Aseptic loosening due to inflammatory osteolysis that occurs in periprosthetic tissue is the leading cause of prosthesis failure and revision surgery. The increasing number of revision surgeries in aged and younger patients clearly points to the need for a new therapeutic intervention to extend the lifespan of a prosthesis[1,2].

Local inflammatory responses triggered by the production of particulate debris released from materials derived from prosthetic components is thought to be the key event associated with periprosthetic osteolysis and aseptic loosening[2]. The generation of wear particles occurs when the articulating surfaces of a prosthesis move across each other under the load of body weight. They initiate biological reactions typified by the activation of damage-associated molecular patterns, Toll-like receptor signaling, NALP3 inflammasomes, and nuclear factor κB (NFκB), resulting in the production of pro-inflammatory cytokines, including interleukin 1 beta (IL-1β), IL-18, tumor necrosis factor alpha (TNF-α), and C-C motif chemokine ligand 20 (CCL20)[2,3]. This facilitates the recruitment of other immune cells resulting in the development of granulomatous chronic inflammation and the formation of synovial-like pseudomembrane around prostheses. Macrophages are believed to play a vital role in the pathogenesis of osteolysis, as they are the predominant cells at the site of periprosthetic tissues, a major source of inflammatory cytokines and can be differentiated into bone-resorbing osteoclasts[2]. However, different cell types such as fibroblasts, dendritic cells, and neutrophils have been also implicated in the pathogenesis of osteolysis[1,4–6]. Consequently, the chronic inflammatory environment at the site of an implant negatively affects bone metabolism and promotes osteoclastogenesis and bone resorption resulting in a loss of implant fixation[2].

Therapeutic targets that reduce osteoclastic-bone resorbing activity or the development of inflammation, including bisphosphonates, a monoclonal antibody to the receptor activator of the NFκB ligand (RANKL), and antibodies to pro-inflammatory cytokines (TNF-α and IL-1β) have failed to prevent pathological bone loss or prolong the lifespan of implants[1,2]. This points to the need for developing an effective therapy for this health problem. Nonetheless, there is a growing body of evidence to suggest that controlling chronic inflammation at the site of an implant would be a promising approach for therapeutic intervention[7]. Chronic inflammation generally occurs when the initial acute inflammation is not effectively resolved due to the inadequate pro-resolving activity of the immune system, a process that is referred to as frustrated resolution. The resolution of inflammation is an active process that is rigidly orchestrated by endogenous pro-resolving mediators that function not as immunosuppressive agents, but instead they promote the resolution of inflammation though activating homeostatic control mechanisms in the affected tissues[8–10]. Of these molecules, Annexin A1 (AnxA1), a member of the annexin superfamily, is mainly released by monocytes and neutrophils and has been implicated in a number of biological processes, including inflammation, intracellular vesicle trafficking, leukocyte migration, tissue growth and regeneration, and apoptosis[11]. In fact, the ability of AnxA1 to stimulate endogenous pro-resolving pathways leading to tissue repair and healing and its therapeutic effects have been documented in a broad range of experimental models, including myocardial ischemia injury, stroke, sepsis, arthritis, and multiple sclerosis[12].

Given that periprosthetic osteolysis is a chronic inflammatory disorder typified by persistent inflammation, and that pro-resolving mediators may restore tissue homeostasis, we explored the function of AnxA1 in the pathophysiology of the disease and evaluated its therapeutic applications in experimental periprosthetic osteolysis models. Our results demonstrated that AnxA1 represents a potential therapeutic agent for periprosthetic osteolysis and other pathological bone resorption-related diseases.

## Results

**Detection of AnxA1 in periprosthetic tissues around osteolytic bone lesions.** To explore the role of AnxA1 in periprosthetic osteolysis, we first examined the issue of whether AnxA1 is present in synovium-like tissues (pseudocapsule) and synovial fluids obtained from three patients who were undergoing revision surgery of total hip arthroplasty. It is noteworthy that AnxA1 was detected in both CD68+ macrophages and Elastase+ neutrophils obtained from inflammatory areas within pseudocapsule tissues as well as in synovial fluids collected from patients who were undergoing revision surgery due to aseptic loosening (Fig. 1a). The CD68+ macrophages appeared to be abundant in periprosthetic tissues and only a few Elastase+ neutrophils were detected in these tissues (Fig. 1a). Importantly, the majority of AnxA1+ cells were detected in the lining layer of pseudocapsule tissues (Supplementary Fig. 1). Consistent with these observations, AnxA1 was detected in CD68+ macrophages and Elastase+ neutrophils that had infiltrated into calvarial tissues in a debris-induced osteolysis mouse model (Supplementary Fig. 1). To further assess that the activation of cells by implant wear particles is associated with an increased production of AnxA1, human differentiated macrophages, freshly isolated neutrophils, and fibroblast-like synoviocytes (hFLS) were stimulated by treatment with ultra-high molecular weight polyethylene (UHMWPE) debris and then subjected Western blot analysis. The expression of AnxA1 did not appear to be significantly increased in macrophages or hFLS stimulated with UHMWPE debris (Fig. 1c, d). Given that resident macrophages are heterogenous and that different phenotypes may be present in periprosthetic tissues, the expression of AnxA1 was examined in differentiated inflammatory and anti-inflammatory macrophages. An increase in the expression of AnxA1 was found in anti-inflammatory macrophages (M2) that had been stimulated with IL-4 (Fig. 1e). Consistent with the observed immunohistochemistry staining of clinical samples, neutrophils that had been stimulated with UHMWPE debris exhibited a significant increase in the expression of AnxA1, but not anti-inflammatory molecules (Fig. 1f, g). AnxA1 was further detected in extracellular vehicles of neutrophils (N-EVs) that had been stimulated with UHMWPE debris (Fig. 1h). To confirm the findings suggesting that neutrophils are active in the presence of implant wear debris, freshly isolated human neutrophils were stimulated with ultra-high molecular weight polyethylene (UHMWPE) debris for 2 h and then subjected to morphological examination by transmission electron microscopy (TEM). Interestingly, the TEM examination demonstrated that the stimulated neutrophils exhibited morphologic characteristics that were typical of activated phagocytes such as cell extension, the presence of cytoplasmic azurophilic granules and large phagolysosomes (Fig. 1i). Apoptotic neutrophils were also present, as evidenced by characteristic chromatin condensation (Fig. 1i). These results suggest that AnxA1 is present in periprosthetic tissues, expressed by macrophages and neutrophils and may have a role in the inflammation and pathological bone resorption induced by wear debris.

**AnxA1 is a potential inhibitor of inflammation and pathological bone resorption induced by wear debris.** To test the supposition that AnxA1 may have an inhibitory role in the pathophysiology of periprosthetic osteolysis, we attempted to manipulate the local expression level of AnxA1 by employing the depletion and adoptive transfer of neutrophils in a murine debris-

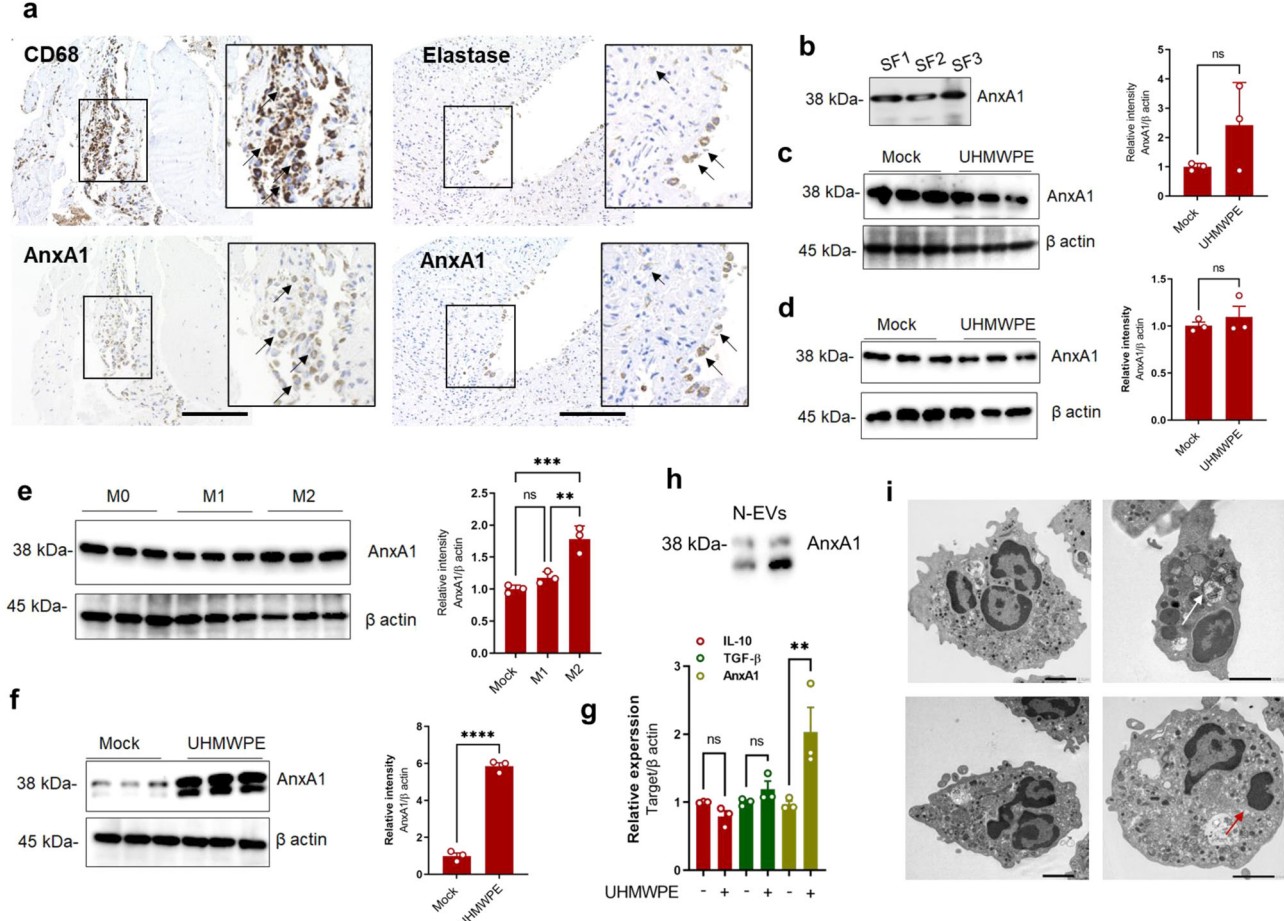

**Fig. 1 Detection of AnxA1 in periprosthetic tissue and the identification of cells expressing AnxA1. a** Detection of AnxA1 in synovial tissues from patients who were undergoing revision surgery by IHC staining with specific antibodies targeting AnxA1, CD68 macrophages and neutrophil elastase. Arrows indicate the CD68+AnxA1+ (left panel) or Elastase +AnxA1+ cells. Scale bars are 100 µm. **b** Detection of AnxA1 in synovial fluids from patients who were undergoing revision surgery by Western blot analysis. **c** Expression of AnxA1 in UHMWPE debris-stimulated macrophages as detected by Western blotting. Right panel shows quantification of band density. **d** Expression of AnxA1 in UHMWPE debris-stimulated hFLS. Right panel shows quantification of band intensity. **e** Expression of AnxA1 in different macrophage phenotypes, including in vitro-generated inflammatory M1 (LPS + IFN-γ) and anti-inflammatory M2 (IL-4) cells. Right panel represent quantification of band intensity. **f** Detection of AnxA1 in UHMWPE debris-stimulated neutrophils. Right panel represents quantification of band intensity. Results represent the mean for the relative band intensity values ± SEM of triplicates. Significant difference between the two groups was determined by two-tailed Student $t$-test and for multiple groups by one-way ANOVA, followed by Tukey's multiple-comparison procedure. *$p < 0.05$, **$p < 0.001$, ***$p < 0.0001$ and $p < 0.00001$. ns indicates not significantly different. **g** Gene expressions of AnxA1 and anti-inflammatory cytokines in UHMWPE debris-stimulated human neutrophils analyzed by qRT-PCR. Results represent the mean of relative expression values ± SEM of three mice. A significant difference between the two groups was determined by the two-tailed Student's $t$ test. ** $p < 0.001$. **h** Detection of AnxA1 in extracellular vesicles of debris-stimulated neutrophils (N-EVs). **i** Morphologic characteristics of UHMWPE debris-stimulated neutrophils assayed by TEM. White arrow indicates phagocytosis and red arrow shows chromatin condensation. Scale bar is 2 µm. Source data are provided as a Source Data file.

induced osteolysis model. A Ly6G monoclonal antibody was employed to deplete the neutrophils in a polyethylene debris-induced osteolysis model. The neutrophil-depleted mice exhibited a significant decline in the expression of AnxA1 in calvarial bone tissue (Fig. 2a, b). It should also be noted that micro-CT revealed that the neutrophil-depleted mice developed significantly greater osteolytic lesions compared to the sham and control mice (Fig. 2c). Histologically, these mice showed a significant increase in TRAP-positive areas and inflammatory infiltrates in calvarial bones (Fig. 2d–f). These increases were accompanied by a significant elevation in the expression osteoclast differentiation makers in calvarial bone tissues (Supplementary Fig. 2). Furthermore, bone marrow-derived neutrophils were adoptively transferred onto calvarial bones following the implantation of the UHMWPE debris. The osteolytic lesions and TRAP-stained

regions in these mice were significantly reduced compared to control mice that had received a PBS injection (Fig. 2g–k). Consistent with the histological observations, significant reductions in the expression of inflammatory cytokines, including *Tnf-α*, *Il-1β*, and *Il-6* in granulomatous tissue that was formed around UHMWPE particles were noted in neutrophil-treated mice (Supplementary Fig. 3). To further confirm these findings, AnxA1-deficient mice were generated and the osteolytic lesions in calvarial bone were evaluated after the implantation of UHMWPE debris. Remarkably, AnxA1-deficient mice (*AnxA1* KO) exhibited significantly greater osteolytic lesions with larger TRAP-stained regions in calvarial bone tissues than those in wild-type mice (Fig. 3a–d). Consistent with these results, these mice exhibited significant elevations in the expression of inflammatory cytokines, including *Tnf-α*, *Il-1β*, and *Il-6* in granulomatous tissue

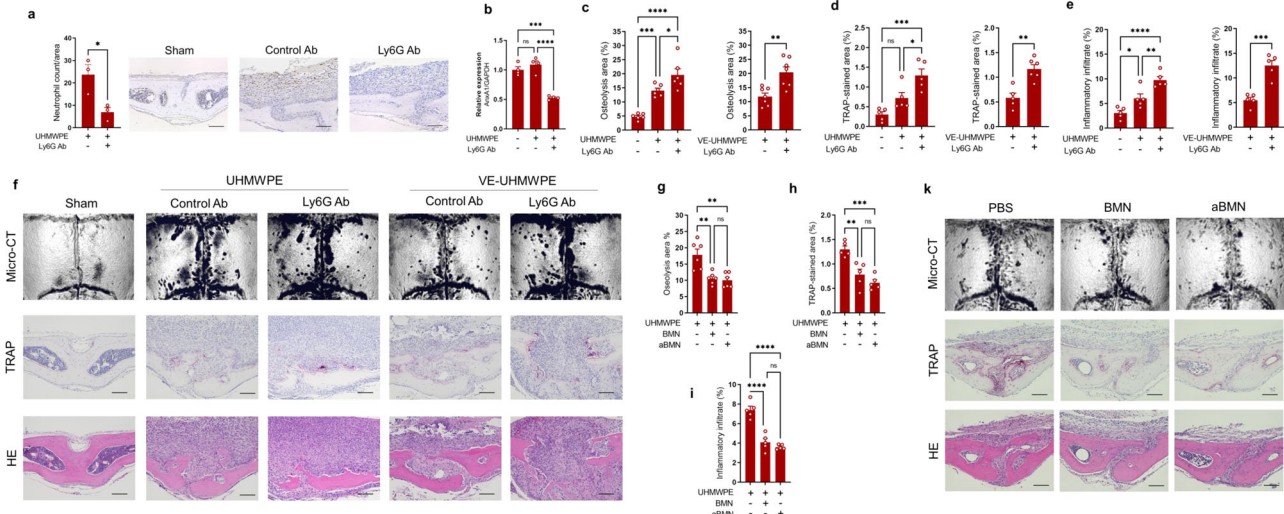

**Fig. 2 Potential inhibitory role of AnxA1 in inflammatory osteolysis induced by implant debris. a** Neutrophil count in Ly6G antibody- and control antibody-treated mice. Detection of neutrophils using IHC in granulomatous tissue formed on calvarial bone after the implantation of UHMWPE debris and treatment by antibodies. Treatment with a Ly6G antibody reduced the number of Elastase positive cells. Results represent the mean of relative expression values ± SEM of 3 mice. Significant difference between the two groups was determined by two-tailed Student's *t* test. Scale bars are 100 μm. **b** AnxA1 expression in calvarial bone tissues of neutrophil-depleted mice analyzed by qRT-PCR. Results represent the mean of relative expression values ± SEM of 4 mice. **c–f** Neutrophil depletion exaggerates the osteolytic activity triggered by UHMWPE and VE-UHMWPE polyethylene debris in a murine model. **c** Quantification of the lytic area in calvarial bone tissues analyzed by micro-CT. Results represent the mean ± SEM of 6 mice. **d, e** Quantification of TRAP-stained areas and inflammatory infiltrate in calvarial bone sections. Significant difference between the two groups was determined by two-tailed Student's *t* test and for multiple groups by one-way ANOVA, followed by Tukey's multiple-comparison procedure. **f** Representative images for micro-CT and histological observations of bone sections stained by TRAP and H&E. Scale bar 100 μm. **g–k** Adoptive transfer of bone marrow neutrophils (BMN) suppress the osteolytic activity triggered by UHMWPE debris. BMN were isolated from donor mice and then injected into calvarial bone after the implantation of UHMWPE debris. Transferred cells were non-stimulated (BMN) or stimulated with TNF-α (aBMN). **g** Quantification of the lytic area in calvarial bone tissues analyzed by micro-CT. **h, i** Quantification of TRAP-stained areas and inflammatory infiltrate in calvarial bone sections. Results represent the means of values ± SEM of 6 mice. Significant difference among groups was determined by one-way ANOVA, followed by Tukey's multiple-comparison procedure. **k** Representative images for micro-CT and histological observations of bone sections stained by TRAP and H&E. Scale bar 100 μm. \*p < 0.05, \*\*p < 0.001, \*\*\*p < 0.0001, \*\*\*\*p < 0.00001. ns indicates no significant difference. Source data are provided as a Source Data file.

that was formed around the UHMWPE debris (Supplementary Fig. 4). Given that AnxA1 is present as a secreted protein or within EVs, we next examined the effects of function-blocking AnxA1 by a specific antibody, which targets secreted AnxA1, on the development of osteolytic lesions in the UHMWPE debris-induced osteolysis model. The local administration of an AnxA1 neutralizing antibody resulted in a significant increase in osteolytic lesions in calvarial bone compared to that in control mice after the implantation of the UHMWPE debris (Fig. 3e–h). These results suggest that secreted AnxA1 acts as a potential inhibitor of inflammation and pathological bone resorption in periprosthetic tissues.

**AnxA1 regulates inflammation and bone resorption through repressing the activation of the NFκB pathway and promoting PPAR-γ signaling pathway.** To gain additional insights into the mechanism responsible for how AnxA1 regulates inflammation and pathological bone resorption, we examined the effects of AnxA1 on transcription factors that are involved in the development of inflammatory responses and the differentiation of osteoclasts in human macrophage cultures. Recombinant human AnxA1 at a concentration of 100 ng/mL, suppressed the expression of phosphorylated NFκB1 (p105/50), NFκB (P65), ERK1/2, but not RelB and P38 in macrophages that had been stimulated by recombinant TNF-α and UHMWPE debris for 24 h (Fig. 4a and Supplementary Fig. 5). The AnxA1 treatment also significantly reduced the number of TRAP-positive cells in cultured macrophages that had been stimulated with UHMWPE debris for 6 days (Fig. 4b). More interestingly, the addition of AnxA1 to RANKL-

stimulated human monocytes resulted in a significant decrease in the number of TRAP-positive cells and areas of bone resorption on dentine slices (Fig. 4c, d). These collective results suggest that Anxa1 acts as a potent inhibitor of the osteoclastogenesis that is triggered by polyethylene wear debris derived from orthopedic implants. To further understand the mechanism by which AnxA1 inhibits osteoclast differentiation, human monocytes were collected from three healthy donors, stimulated by RANKL with or without AnxA1 and then subjected to RNA sequencing. A bioinformatic analysis revealed that AnxA1 reduced the expression of genes clustered in osteoclast differentiation and the NFκB pathway and enhanced the expression of genes clustered in the PPAR-γ pathway (Fig. 4e). Of interest, *PPAR-γ* was in the list of top-regulated genes with a fold change of 7.16 (Fig. 4f). The increased expression of PPAR-γ was confirmed by Western blot analysis that showed a significant elevation in PPAR-γ expression after stimulating macrophages with AnxA1 for 180 min (Fig. 4g). Bearing in mind that formyl peptide receptor 2 (FPR2) is the main receptor of AnxA1 that conveys its biological functions, we examined the increased expression of PPAR-γ in AnxA1 stimulated macrophages that had been pretreated with a selective antagonist of FPR2 (WRW4). Of note, the ability of AnxA1 to promote the expression of PPAR-γ was slightly reduced in macrophages that had been pretreated with WRW4 (Supplementary Fig. 6). These collective results suggest that AnxA1 may function as a potential suppressor of inflammation and pathological bone resorption through activating PPAR-γ pathway.

Given the vital role of fibroblasts and osteoblasts in the development of periprosthetic osteolysis triggered by particulate

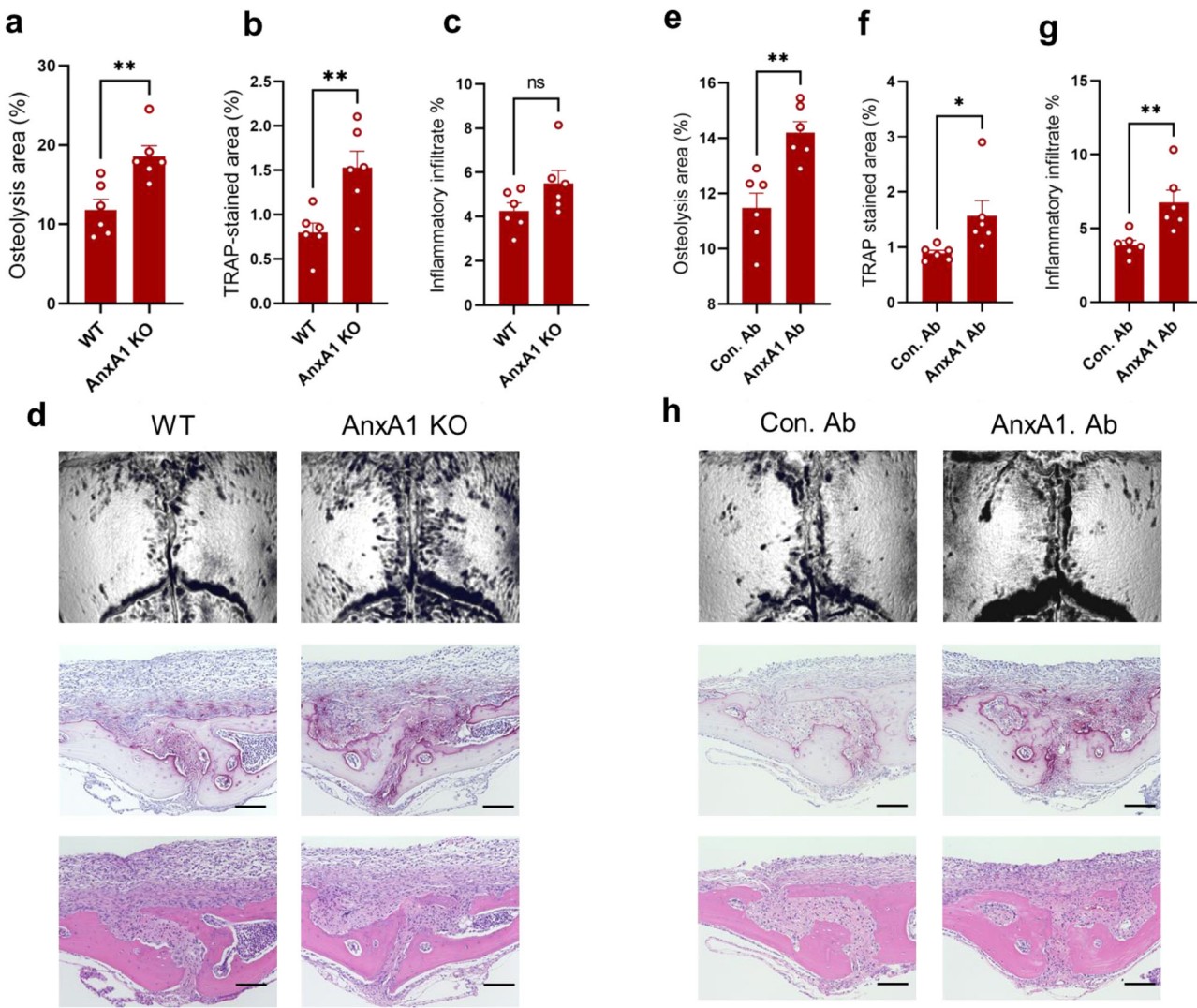

**Fig. 3 Inhibitory role of AnxA1 in inflammatory osteolysis induced by implant debris. a** Quantification of the lytic area in calvarial bone tissues of wild-type WT and AnxA1-deficient mice after implantation of UHMWPE as analyzed by micro-CT. Results represent the mean ± SEM of 6 mice. **b**, **c** Quantification of TRAP-stained areas and inflammatory infiltrate in calvarial bone sections. Results represent the mean ± SEM of 6 mice. **d** Representative images for micro-CT and histological observations of bone sections stained by TRAP and H&E. Scale bar 100 µm. **e** Quantification of the lytic area in calvarial bone tissues of mice that received a function-blocking antibody (AnxA1. Ab) or a control (Con. Ab) analyzed by micro-CT. **f**, **g** Quantification of TRAP-stained areas and inflammatory infiltrate in calvarial bone sections of Ab-treated mice. Results represent the mean ± SEM of 6 mice. **h** Representative images for micro-CT and histological observations of bone sections stained by TRAP and H&E. Scale bar 100 µm Significant difference between the two groups was determined by two-tailed Student's *t* test. *$p < 0.05$, **$p < 0.001$. ns indicates no significant difference. Source data are provided as a Source Data file.

wear debris, we further evaluated the effects of AnxA1 on these cells in vitro. It is noteworthy that the expression of inflammatory mediators in synoviocytes that had been stimulated with TNF-α were significantly suppressed by AnxA1 (Supplementary Fig. 7a). In addition, the Anxa1 treatment significantly enhanced the expression of the osteoblast-anabolic factors in osteoblasts that had been stimulated with TNF-α (Supplementary Fig. 7b). These results demonstrate that AnxA1 exerts beneficial effects on a variety of cell types at the inflammation sites.

**Pharmacological potential of AnxA1 for treating periprosthetic osteolysis and pathological bone resorption**. Given the fact that the AnxA1 N-terminal derived peptide Ac2–26 exerts the same pro-resolving properties as the full-length AnxA1 in numerous inflammatory disease models, we assessed the therapeutic effect of the mimetic peptide Ac2–26 for treating periprosthetic osteolysis

using murine experimental models. The ability of Ac2–26 to promote the expression of PPAR-γ was first confirmed in THP1 macrophages (Supplementary Figu. 8). In addition, UHMWPE particles were implanted into calvarial bone tissues and Ac2–26 was locally injected for 5 consecutive days. The other two groups of mice were treated with BML111, a selective agonist of the formyl peptide receptor 2 (FPR2) for 5 consecutive days or with WRW4, a selective antagonist of FPR2, on days 1, 3, and 5. Interestingly, the mice that received Ac2–26 showed significantly decreased bone osteolytic lesions, TRAP staining and inflammation compared to control mice that had been injected with vehicle (Fig. 5a–d). In these mice, the gene expression of inflammatory cytokines was significantly reduced in granulomatous tissue around UHMWPE particles (Supplementary Fig. 9). In contrast, neither the BML111 nor the WRW4 treatment altered/exacerbated the osteolytic lesions induced by UHMWPE particles (Fig. 5a–c). Our results lead to the conclusion that AnxA1 N-terminal peptide Ac2–26 may offer a

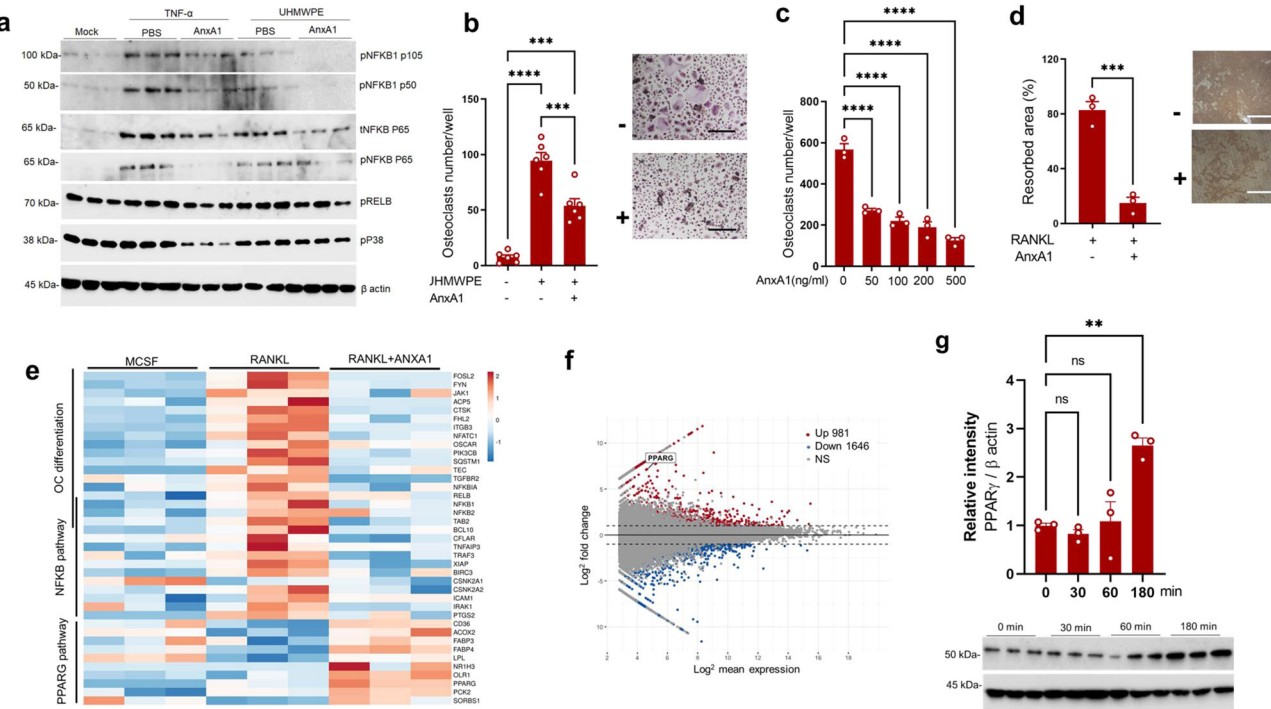

**Fig. 4 Molecular regulatory function of AnxA1 in inflammation and pathological bone resorption. a** Effects of AnxA1 of the expression of transcriptional factors involved in inflammation and osteoclast differentiation. **b** Inhibitory effects of AnxA1 on osteoclast formation in macrophages stimulated with UHMWPE debris. Results represent the mean ± SEM of 6 samples. Right panel shows representative images for stained osteoclasts by TRAP. Scale bars are 100 μm. **c, d** Inhibitory effects of AnxA1 on osteoclast differentiation and bone resorption. Macrophages were stimulated with RANKL in the presence of different concentrations of AnxA1. **d** Quantification of bone resorption area (pitting) formed by RANKL-stimulated macrophages in a 100 ng/mL concentration of AnxA1 in bone resorption assay. Results represent the mean ± SEM of 3 samples. Significant difference between the two groups was determined by two-tailed Student's t test and for multiple groups by one-way ANOVA, followed by Tukey's multiple-comparison procedure. Scale bars are 200 μm. **e** Heat map based on a KEGG pathway enrichment analysis for upregulated genes in macrophages stimulated with RANKL and AnxA1. **f** MA plot analysis for transcript expression levels of significantly up-or down-regulated genes in macrophages stimulated with RANKL and AnxA1 ($p < 0.05$) ($n = 3$). **g** Detection of PPAR-γ by Western blot analysis in macrophages stimulated with AnxA1 at different time points. Left panel shows the quantification of the relative intensity of the detected bands and results represent the mean ± SEM of 3 samples. Significant difference among the groups was determined by one-way ANOVA, followed by Tukey's multiple-comparison procedure. **$p < 0.001$, ***$p < 0.0001$, ****$p < 0.00001$. ns indicates no significant difference. Source data are provided as a Source Data file.

novel therapeutic agent for the treatment of periprosthetic osteolysis triggered by polyethylene particles. To gain further insights into the therapeutic mechanism of Anxa1, mice that had received UHMWPE particle implants were treated with the Ac2–26 mimetic peptide and a PPAR-γ antagonist (GW9662). Remarkably, the therapeutic effects of Ac2–26 were abrogated in mice that had received GW9662 (Fig. 6a–d), suggesting a correlation between the therapeutic mechanism of Anxa1 and the activation of the PPAR-γ pathway. the elevated expression of inflammatory cytokines noted in granulomatous tissue around the UHMWPE particles in GW9662-treated mice was consistently accompanied by a reduction in the expression of PPAR-γ (Fig. 6e). To obtain further evidence for the potential therapeutic applications of AnxA1 in pathological bone resorption-related diseases, TNF-α- and RANKL-induced bone loss models were treated with the Ac2–26 mimetic peptide. Treatment with the Ac2–26 peptide resulted in a significant reduction in bone loss and TRAP-stained areas in both models (Fig. 7a, b). This treatment also significantly suppressed the infiltration of inflammatory cells in the inflammatory osteolysis model induced by TNF-α (Fig. 7a–d). From these results, we conclude that AnxA1 inhibits the pathological bone resorption induced by TNF-α- and RANKL, known as the main osteoclastogenic factors in periprosthetic osteolysis.

Given the clinical relevance of these findings, we developed an Ac2–26-mixed matrigel that allowed the locally controlled release

of the peptide onto the calvarial bone and evaluated this in the TNF-induced bone loss model. Matrigel is a natural hydrogel that has been successfully used to study cell migration, angiogenesis, and protein/peptide delivery. The peptide was formulated into a thermo-responsive material that is soluble at lower temperature but turns into a semi-solid hydrogel in situ in response to the normal body temperature (37 °C) (Fig. 8a). The desired formulation would be expected to produce a controlled-release pattern for the peptide through a time span of one week and to therefore result in a sustained and prolonged effect (Fig. 8a, b). Of note, a single administration of the Ac2–26-mixed Matrigel onto calvariae (beyond lambdoid suture) alleviated osteolytic lesions and the pathological bone resorption induced by the TNF-α administrations (Fig. 8c–g). The results obtained from the Matrigel treatment were comparable to the results for treatment with the mimetic peptide Ac2–26 for 4 consecutive days. These collective data demonstrate the potential therapeutic and translatable applications of AnxA1 in periprosthetic osteolysis and pathological bone resorption-related diseases.

## Discussion
Promoting the resolution of inflammation that is associated with periprosthetic osteolysis represents a promising treatment strategy, since this process is a fundamental mechanism for restoring

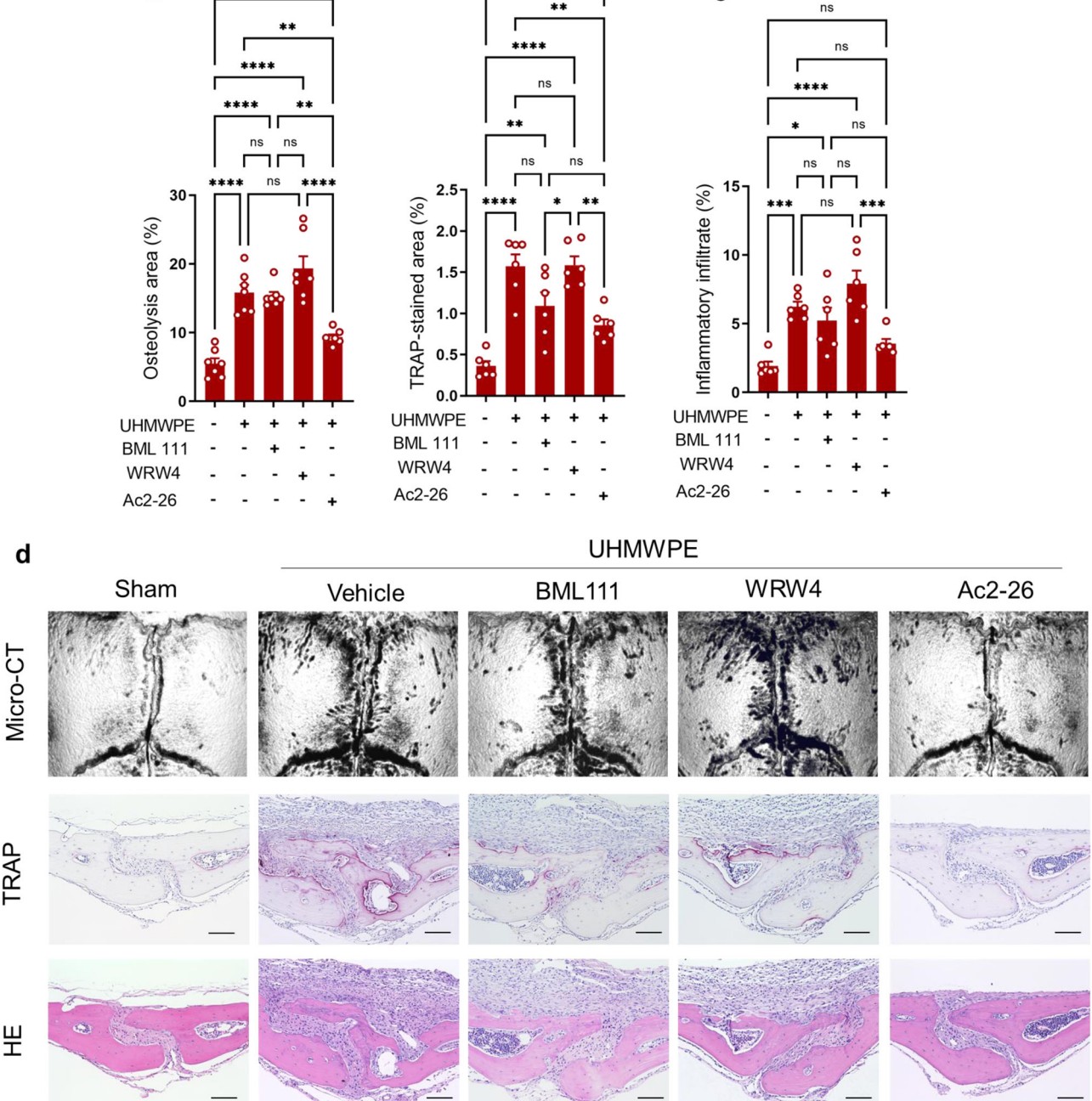

**Fig. 5 Therapeutic effect of Ac2–26 in UHMWPE debris-induced inflammatory osteolysis model.** Debris was implanted and the calvarial bone of mice were then treated with BML111 (1 mg/kg), WRW4 (2.5 mg/kg) and Ac2–26 (1 mg/kg). **a** Quantification of the lytic area in calvarial bone tissues analyzed by micro-CT. Results represent the mean ± SEM of 7 mice. **b**, **c** Quantification of TRAP-stained areas and inflammatory infiltrate in calvarial bone tissues. Significant difference among the groups was determined by one-way ANOVA, followed by Tukey's multiple-comparison procedure. $*p < 0.05$, $**p < 0.001$, $***p < 0.0001$, $****p < 0.00001$. ns indicates no significant difference. **d** Representative images for micro-CT and histological observations of bone sections stained by TRAP and H&E. Scale bar 100 μm. Source data are provided as a Source Data file.

tissue homeostasis[8,9]. Therefore, we explored the pro-resolving function of AnxA1 in periprosthetic osteolysis and assessed its therapeutic potential in experimental periprosthetic osteolysis models.

AnxA1 was expressed at high levels in macrophages and neutrophils within the tissues around the loosening implant. Our further results demonstrated that the deletion of or the blocking of AnxA1 resulted in the development of severe osteolytic lesions in a murine wear debris-induced osteolysis model, suggesting that

AnxA1 has a protective function in inflammatory osteolysis. These conclusions are supported by the in vitro findings indicating that recombinant AnxA1 exhibited inhibitory effects on the inflammatory responses of macrophages and RANKL-induced osteoclast differentiation through reducing the expression of NFκB transcription factors. These findings appear to be consistent with findings reported in an earlier study showing the suppressive activity of AnxA1 on inflammation and osteoclast formation in an arthritis model[13]. It is noteworthy that

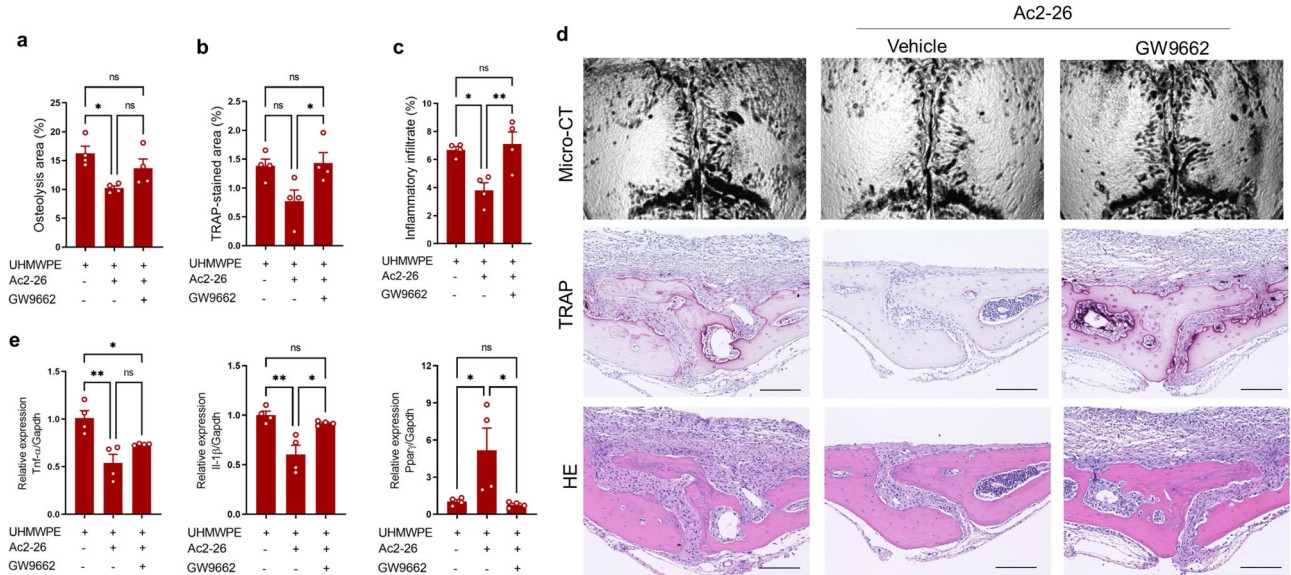

**Fig. 6 Treatment with aPPAR-γ antagonist impaired the therapeutic effect of Ac2–26 in a UHMWPE debris-induced inflammatory osteolysis model.** Debris was implanted and mice were then treated by a daily local injection of AnxA1 (1 mg/kg) and an intraperitoneal injection of GW9662 (1 mg/kg). **a** Quantification of the lytic area in calvarial bone tissues analyzed by micro-CT. **b**, **c** Quantification of TRAP-stained areas and inflammatory infiltrate in calvarial bone tissue. **d** Representative images for micro-CT and histological observations of bone tissues stained by TRAP and H&E. Scale bars 100 μm. **e** Gene expression of inflammatory molecules in granulomatous tissue that was formed around UHMWPE debris as analyzed by qRT-PCR. Results represent the mean ± SEM of 4 mice. Significant difference among the groups was determined by one-way ANOVA, followed by Tukey's multiple-comparison procedure. *$p < 0.05$, **$p < 0.001$. ns indicates no significant difference. Source data are provided as a Source Data file.

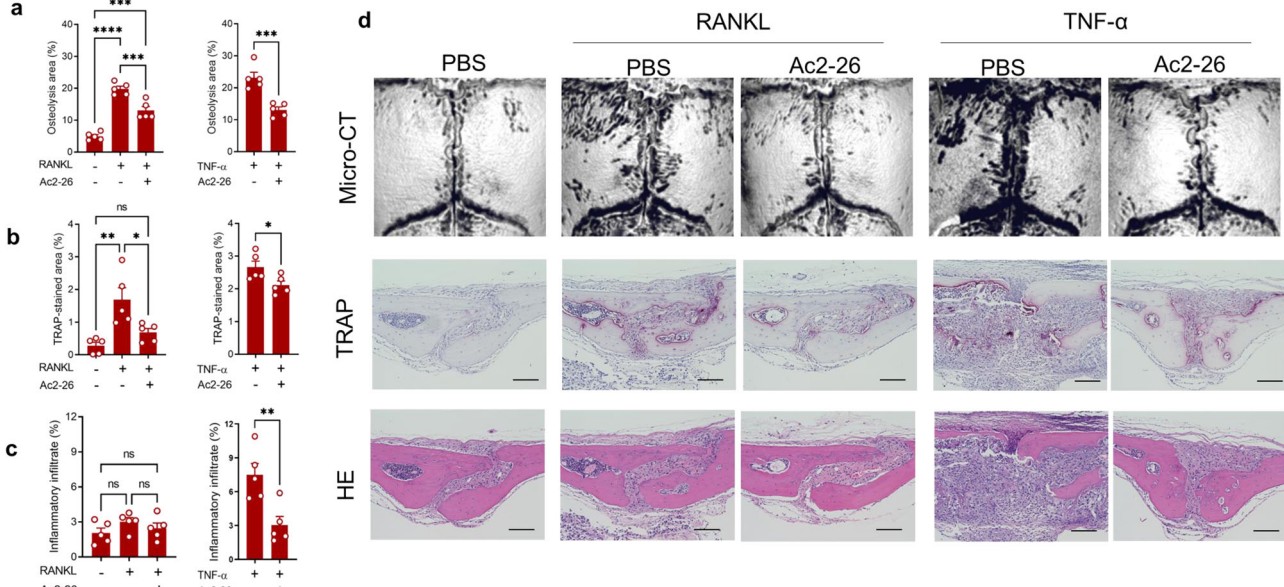

**Fig. 7 Therapeutic effect of Ac2–26 on pathological bone loss induced by TNF-α and RANKL administration.** Calvarial bones were locally injected with cytokines for 4 days and bone erosion was then quantified on day 5. **a** Quantification of the lytic area in calvarial bone tissues analyzed by micro-CT. Results represent the mean ± SEM of 5 mice. **b**, **c** Quantification of TRAP-stained areas and inflammatory infiltrate in calvarial bone sections. Significant difference between the two groups was determined by two-tailed Student's $t$ test and for multiple groups by one-way ANOVA, followed by Tukey's multiple-comparison procedure, and was shown as *$p < 0.05$, **$p < 0.001$, ***$p < 0.0001$, ****$p < 0.00001$. ns indicates no significant difference. **d** Representative images for micro-CT and histological observations of bone sections stained by TRAP and H&E. Scale bar 100 μm. Source data are provided as a Source Data file.

macrophages that had been treated with AnxA1 exhibited an elevation in the expression of PPAR-γ and a reduction in the levels of NFκB transcription factors. In line with our findings, the activation of PPAR-γ has been reported to be a promising approach for the treatment of inflammatory diseases and cancer through reducing NFκB p65 transcriptional activity[14,15]. In addition, the activation of PPAR-γ drives the polarization of monocytes to M2 macrophages with anti-inflammatory properties and inhibits RANKL- and TNF-α-mediated osteoclast differentiation[15–17]. In a related study, IL-4 was reported to inhibit RANKL-induced osteoclast formation though the activation of PPAR-γ and suppressing the activation of the NFκB

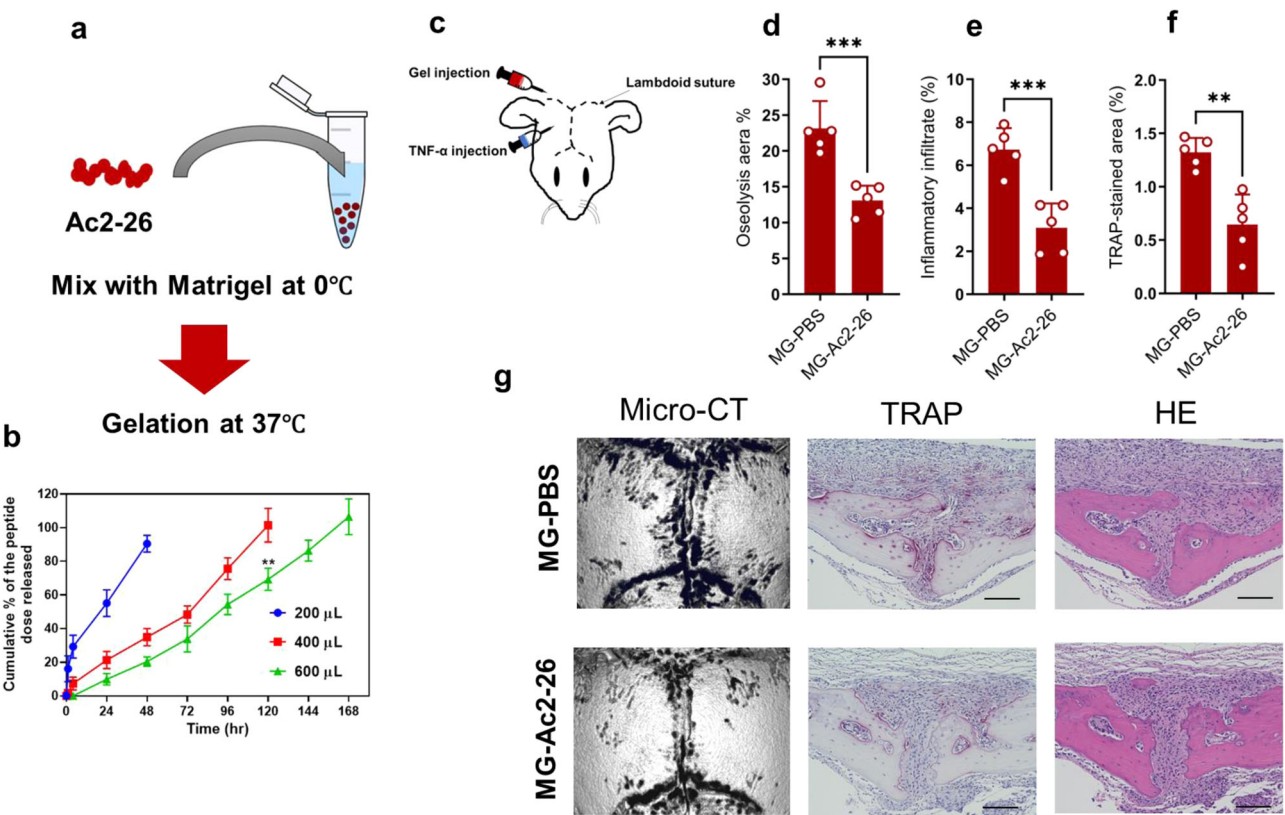

**Fig. 8 Matrigel encapsulating Ac2–26 and its pharmacological potential for the treatment of pathological bone resorption induced by TNF-α.**
**a** Schematic diagram showing the procedure for preparing the Matrigel encapsulating the Ac2–26 peptide. **b** In vitro release of the peptide from hydrogels containing different amounts of Matrigel at 37 °C and pH 7.4. The results represent the mean ± SEM of independent experiments. **p < 0.001 Significance was determined by the two-tailed Student's t test. **c** Schematic diagram for the injection of gel and cytokines. **d–g** Therapeutic effect of Matrigel encapsulating Ac2–26 in a bone loss model induced by TNF-α administration. The MG-PBS was the control gel, and the MG-Ac2–26 is test gel. **d** Quantification of lytic area in calvarial bone tissues analyzed by micro-CT. **e, f** Quantification of TRAP-stained areas and inflammatory infiltrate in calvarial bone tissues. A significant difference between groups was determined by the two-tailed Student's t test. *p < 0.05, **p < 0.001, ***p < 0.0001. Results represent the mean ± SEM of 5 mice. **g** Representative images for micro-CT and histological observations of bone sections stained by TRAP and H&E. Scale bar 100 μm. Source data are provided as a Source Data file.

pathway[18]. Therefore, the possible inhibitory effects of AnxA1 could be attributed to its ability to decrease the activation of NFκB signaling, known as a positive regulator of the pro-inflammatory function of macrophages and the bone-resorbing function of osteoclasts. Inhibition and gene deletions experiments revealed that the activation of NFκB is a central mediator of chronic inflammatory diseases, including inflammatory bowel diseases, rheumatoid arthritis, and asthma. The canonical NFκB pathway includes the activation of two catalytic subunits of IKKα and IKKβ, and the regulatory subunit of IKKγ (NEMO). IKK phosphorylates IκBα, leading to IκBα degradation in proteasomes, resulting in the rapid nuclear translocation of inducible transcription factors including NF-κB1 (p100/50), NF-κB2 (p52), RelA (p65), RelB and c-Rel, which are associated with the induction of a large number of inflammatory genes. Similarly, IKKβ activity is involved in RANKL signaling in osteoclast precursors, leading to the differentiation of macrophages into osteoclasts[19]. Gene manipulation experiments highlight the essential role of IKKβ in the development of inflammatory and osteoclastogenesis. Therefore, NF-κB inhibitors, namely these targeting the IKK protein family, have shown great potential for treating inflammatory diseases and bone diseases that are associated with impaired bone-remodeling processes such as rheumatoid arthritis and inflammatory osteolysis[20–22].

The therapeutic effects of AnxA1 against inflammatory osteolysis were evident both in vitro and in vivo models. Our

results show that AnxA1 suppressed local inflammatory responses, immune cell infiltration, and bone resorbing osteoclast activity in calvarial osteolysis models. AnxA1 is abundantly present in inflammatory exudates and is known to act as an anti-inflammatory mediator through activating the pro-resolving phase receptor FPR2. AnxA1 is a member of the annexin superfamily of calcium-dependent phospholipid-binding proteins that is abundantly expressed in neutrophils and other innate immune cells and functions as an inflammation-resolving molecule through directing leukocyte phagocytosis, differentiation, migration, and apoptosis. AnxA1 is able to turn on c-Jun N-terminal kinase (JNK)-signaling, leading to the production of IL-10 and can also enhance TGF-β signaling via promoting Smad activity[23–25]. These actions result in a decrease in the production of inflammatory cytokines and redirects the polarization of macrophages towards an anti-inflammatory M2 phenotype, favoring a resolving/repair phase in the tissue[26]. In a related study, in 2020, McArthur et al. reported that the AnxA1/AMPK axis as an important pathway for inducing the production of the pro-resolving macrophage phenotype required for skeletal muscle injury regeneration[26]. In support of this concept, AnxA1 knockout mice show severe pathological damage in affected tissues that are associated with increased inflammatory responses and high levels of phosphorylated ERK-1/2 and NF-κB p65[27]. It is also noteworthy that PPAR-γ activation by rosiglitazone exerts pharmacological effects against alcoholic fatty livers in mice by

modulating the activity of AMPK, suggesting the existence of a correlation between PPAR-γ and AMPK signaling[28]. Consistent with this view, resolvin D1, the activation of a related pro-resolving molecule, inhibits inflammation and the NF-κB signaling pathway in a mechanism that is dependent on PPAR-γ activation[29]. These findings provide an explanation for the loss of the therapeutic effects of AnxA1 after injecting a PPAR-γ antagonist in a wear debris-induced osteolysis model. Our collective data indicate that the AnxA1/PPAR-γ signaling pathway may a play regulatory role in periprosthetic osteolysis by attenuating the inflammation and pathological bone resorption induced by implant wear debris.

The therapeutic potentials of AnxA1 and its peptide Ac2–26 have also been extensively demonstrated in experimental models, such as in allergic conjunctivitis, and in ischemic stroke models[30–32]. The pharmacological properties of AnxA1 and its peptide Ac2–26 are known to be dependent on their ability to activate FPR2 signaling which makes it a promising anti-inflammatory and pro-resolving therapeutic target for controlling chronic inflammatory diseases. However, there is a line of evidence to suggest that Ac2–26 inhibits cell adhesiveness and migration via downmodulating α4β1 integrin and their affinity and valency, without changing their cell surface expression[30]. Likewise, AnxA1 protects the cerebrum from thromboinflammation through reducing the levels of pro-inflammatory cytokines and regulating the adhesion/aggregation of leukocytes and platelets to cerebral microvascular endothelial cells[31].Therefore, it is possible that the therapeutic effects of Ac2–26 in inflammatory osteolysis might be due to its ability to reduce integrin-dependent monocyte adhesion and the migration necessary for the development of inflammation and osteoclast formation[33]. In line with this supposition, neither the activation of FPR2 by an agonist treatment (BML111) nor blocking by an antagonist treatment (WRW4) had a significant impact on the pathological bone lesions induced by debris implantation, suggesting that the therapeutic potentials of Ac2–26 is independent of FPR2 signaling. Our promising findings regarding the potential therapeutic use of AnxA1 have stimulated the development of controlled-release hydrogels containing the Ac2–26 mimetic peptide. A nanomedicine approach has the potential for clinical translation, since it protects short peptides from proteolysis in vivo and facilitates the delivery of the cargo to the injury site without the need to repeat the injection. Treatment with Matrigel containing Ac2–26 caused a significant reduction in pathological bone resorption and inflammation. The obtained results were comparable to the results for the mimetic peptide treatment, which confirmed the use of Ac2–26 in the treatment of periprosthetic osteolysis and pathological bone resorption-related diseases. A similar approach was successfully developed for dermal wound repair applications with an optimal controlled release formulation of Ac2–26[34]. Moreover, it has been reported that nanoparticles that allow the slow release of Ac2–26 reduce tissue damage in zymosan-induced peritonitis and protect against advanced atherosclerosis in hypercholesterolemic mice[35,36]. Further studies including the development of safer nanomaterials for the delivery of Ac2–26 promise to have great potential in clinical applications.

In conclusion, the present study sheds light on a new strategy for innovative therapeutic approach for preventing implant failure. The AnxA1/PPAR-γ axis appears to play an important role in attenuating inflammation and pathological bone resorption in periprosthetic osteolysis. Our study highlights AnxA1 as clinically translatable therapeutic agent for the prevention of implant loosening.

## Methods

**Study approval**. The research protocols for human samples were approved by the Research Ethics Review Committee of Hokkaido University Hospital (Approval ID:

016-0002), and informed consents were obtained from all donors. Procedures for animal experiments were performed in accordance with our approved protocols (nos. 17-0085 & 18-0171) by the Institute of Animal Care and Use Committee of the Hokkaido University Graduate School of Medicine. All mice were kept in SPF condition and were maintained in a controlled temperature of 21–22 °C, humidity of 40 and 50% and a constant 12-h light/dark room during the experiments.

**Immunostaining of synovial tissues**. Formalin-fixed synovial tissues from three patients (one male of 60-year-old, two females 54- and 59-year old) undergoing revision of total hip arthroplasty were embedded in paraffin and 3 μm sections were used for immunohistochemistry staining (IHC). All samples didn't show any clinical signs of local infection with the level of serum CRP below 0.07 mg/dl. Control tissue was collected from patient of hip osteoarthritis (female 64-year-old) undergoing primary hip arthroplasty. Sections were deparaffinized, treated for 5 min with proteinase K (Dako, CA, USA) for antigen retrieval, and then blocked with horse serum for 1 h. Sections were incubated with primary antibodies to myelo-peroxidase (MPO: JM10–58, 1:200 dilution, Novus Biologicals, USA), neutrophil Elastase (ab68672, 1:200 dilution, Abcam, Cambridge, UK), CD68 (KP1, clone PG-M1, M0876: 1:200 dilution, Dako Agilent, USA), and AnxA1 (GTX113329; 1:200 dilution, GeneTex, CA, USA) overnight and then washed three times with tris-buffered saline buffer (TBS). The signal was amplified with Vectastain Elite ABC kit for detection of horseradish peroxidase (HRP) (Vector Laboratories, Burlingame, USA) followed by counterstaining with hematoxylin for detecting cellular nuclei.

**Wear debris preparation**. Fabricated wear particles generated from bearing materials of hip implants of virgin UHMWPE or its vitamin E-blended materials (VE-UHMWPE) (Teijin Nakashima medical, Okayama, Japan) were prepared with size range of 0.1–100 μm as described in our earlier studies[3,37]. Briefly, GUR1020 powder (Celanese Japan, Tokyo, Japan) and GUR1050 powder blended with 0.3 wt% dl-α-tocopherol (Eisai, Tokyo, Japan) were subjected to direct compression molding technique to prepare bulk materials of virgin UHMWPE and VE-UHMWPE, respectively. Thereafter, the manufactured bulk materials were crushed by using a Multi Beads Shocker (Yasui Kikai, Osaka, Japan) at 3500 rpm and then sterilized using an ethylene oxide gas (EOG) sterilizer (Eogelk-SA-H160, Osaka, Japan). Equivalent circle diameter for particle sizes was determined using the Image analyzer Morphologi G3 (Malvern Instruments, Worcester, UK). Samples were 0.1–100 μm sizes and endotoxin-free as tested using a ToxinSensor Single Test Kit (Genscript, Piscataway, NJ, USA).

**Neutrophil culture and stimulation**. Human peripheral blood was collected using BD Vacutainer tubes containing ACD (Becton, Dickinson and Company, NJ, USA) from three healthy Asian donors with no history of inflammatory diseases, joint disorders and total joint arthroplasty. Human neutrophils were separated by using Ficoll-Paque™ PLUS (GE Healthcare) and then subjected to a MACSxpress neutrophils isolation kit (Miltenyi Biotec., CA, USA). The purity of isolated cells was confirmed (>97%) by performing a differential blood cell count of Giemsa-stained smears. Freshly isolated neutrophils $1 \times 10^6$ were seeded onto poly-d-lysine-coated wells and stimulated with UHMWPE debris at a density of 0.1 mg/cm$^3$ in minimum essential medium Eagle (MEM, Sigma) supplemented with 10% heat-inactivated fetal bovine serum (FBS, Nichirei Biosciences INC, Tokyo, Japan), and 5% mg/L penicillin/streptomycin solution (Wako, Japan) for 2 h at 37 °C in a humidified atmosphere containing 5% CO$_2$ using inverted culture system[3]. For isolation of extracellular vesicles (EVs) of stimulated neutrophils, freshly isolated neutrophils $4 \times 10^6$ were stimulated with UHMWPE debris, and the cell suspension diluted in PBS (10 mL total sample volume) was centrifuged at 400 g for 5 min. The supernatant was centrifuged at 2000 g for 10 min, followed by passing through 0.45 μm syringe filter (Advantec INC., CA, USA). EVs were then isolated by ultracentrifugation of the supernatant at 100,000 g for 1 h. Pelleted EVs were washed with PBS and centrifuged again at 100000 g for 1 h. All steps were done at 4 °C. The pellet was suspended in 50 μl PBS and then subjected to Western blot analysis. Cells stimulated with recombinant TNF-α (Biolegend, San Diego, USA) were used as a positive control. For transmission electron microscopic examination, cultured neutrophils with UHMWPE debris were washed with cold PBS and fixed in 1.5% formaldehyde and 1% glutaraldehyde in cacodylate buffer with pH 7.4. The cells were then rinsed in cacodylate buffer and resuspended in 1% osmium tetroxide for 2 h, followed by rinsing with cacodylate buffer. Cells were dried in a graded ethanol series followed by embedding in epoxy resin. Sections were incubated with 2% uranyl acetate for 10 min at room temperature, followed by rinsing by DW, and stained for 6 min with Reynolds lead citrate stain (Sigma). Neutrophils were visualized by a Hitachi H-7100 transmission electron microscope at 80 kV (JEM-1400, Plus Nihon Denshi, Tokyo, Japan).

**Macrophage stimulation and osteoclast differentiation, and their function assays**. Human primary monocytes were isolated from the blood samples of healthy donors by density gradient centrifugation (Ficoll-Paque™ PLUS: GE Healthcare, Waukesha, WI, USA) followed by treatment with a MACS Pan monocyte isolation kit (Miltenyi Biotec, Auburn, CA, USA). The differentiation of macrophages and osteoclasts and related function assays were essentially performed as described in previous studies[37,38]. The attached cells were cultured in MEM

supplemented with 10% heat-FCS, 5% mg/l penicillin/streptomycin solution, and 5% L-glutamine in a 37 °C-humidified atmosphere containing 5% CO2 for 3 days in the presence of 25 ng/mL of recombinant macrophage colony-stimulating factor (MCSF, Peprotech, NJ, United States). Differentiated macrophages were stimulated with UHMWPE debris in the presence or absence of 100 μg recombinant human AnxA1 (R&D Systems, MN, USA) using an inverted culture system[3]. Cultured macrophages were subjected to Western blot analysis (24 h stimulation) and TRAP staining (6 days stimulation). Macrophages stimulated with 10 μg/mL recombinant human TNF-α (Peprotech) were used as a positive control. In a separate experiment, differentiated macrophages were stimulated with different concentrations of recombinant human AnxA1 (R&D Systems) and harvested at different time points. In addition, cells from a human monocyte cell line THP1 (RIKEN, Saitama, Japan) were seeded at a density of $1 \times 10^5$ cells/well onto a 48-well plate and allowed to differentiate into macrophages in the presence of 5 ng/mL phorbol myristate acetate (PMA, Sigma) for 48 h. The cells were next stimulated by treatment with recombinant AnxA1 or synthetic peptide Ac2–26 (KareBay™ Biochem, Inc., NJ, USA). For the preparation of macrophage phenotypes, the cells were stimulated for 48 h with a 100 ng/mL solution of lipopolysaccharide (LPS; Sigma) plus a 100 ng/mL solution of recombinant human interferon gamma (IFN-γ) for M1 macrophages and 200 ng/mL IL-4 (Peprotech) for M2 macrophages. For osteoclast differentiation, monocytes were cultured in medium supplemented with 25 ng/mL MCSF and 50 ng/mL RANKL (Peprotech) plus different concentrations of recombinant human AnxA1 (R & D systems). Cells were stained on day 8 using a TRAP kit (Sigma) and TRAP-positive stained cells with ≥3 nuclei were counted as the osteoclasts. Bone pits on dentine slices were detected by staining with 20 mg/mL of a solution of peroxidase-conjugated wheat germ agglutinin and 3,3'-diaminobendzidine (0.52 mg/mL in PBS containing 0.1% $H_2O_2$) on day 21. Slices were examined under a confocal microscope and pits were calculated as the percentage of resorbed bone surface/total bone surface area (ImageJ, National Institutes of Health, NIH, Washington, DC, USA).

**FLS stimulation.** The hFLS from normal healthy human synovial tissues purchased from Cell Applications (Cell Applications) were cultured according to the supplier's recommendations. Cultured cells were stimulated with UHMWPE debris at a density of 0.1 mg/cm³ for 24 h at 37 °C in a humidified atmosphere containing 5% CO₂ using an inverted culture system[3]. In separate experiment, cultured hFLS cells were stimulated with recombinant human TNF-α (Peprotech) in the presence or absence recombinant AnxA1 (R&D Systems).

**Generation of AnxA1-deficient mice (AnxA1 KO).** Generation of AnxA1 KO mice was carried out using CRISPR/Cas9 technique. Briefly, two gRNA targets (5'-GAA CAC CGG TGA TTA CGC TG-3' and 5'-CTG GCA CTC TTG GTC AGA AG-3') located on intron 3 and 5 of Anxa1, respectively, were selected for producing the Anxa1 KO mouse in which exons 3 to 5 of the Anxa1 had been excised. These two gRNAs were synthesized and purified by means of a GeneArt Precision gRNA Synthesis Kit (Thermo Fisher Scientific, Waltham, Massachusetts) and dissolved in Opti-MEM (Thermo Fisher Scientific, Waltham, Massachusetts). Thereafter, the mixture of two gRNAs (25 ng/μL, each) and GeneArt Platinum Cas9 Nuclease (100 ng/μL) were electroplated into the zygotes of C57BL/6 J mice (Charles River Laboratories Japan, Yokohama, Japan) by using the NEPA 21 electroporator (Nepa Gene Co. Ltd., Ichikawa, Japan). After electroporation, two-cell embryos were transferred into the oviducts of pseudopregnant female and newborns were obtained. Mice were kept under SPF conditions. Genotyping was routinely performed using genomic DNA extracted from <0.5 mm tails of 3–4-week-old mice. PCR was carried out using AmpliTaq Gold 360 Master Mix (Thermo Fisher Scientific, Waltham, Massachusetts) with the appropriate primers (Supplementary Table 1). The PCR products were purified with a FastGene Gel/PCR Extraction Kit (Nippon Genetics, Tokyo, Japan) and the DNA sequences were obtained using a BigDye Terminator v3.1 Cycle Sequencing Kit (Thermo Fisher Scientific, Waltham, Massachusetts), FastGene Dye Terminator Removal Kit (Nippon Genetics, Tokyo, Japan), and 3500xL Genetic Analyzer (Thermo Fisher Scientific, Waltham, Massachusetts).

**Calvarial osteolysis models.** Debris-induced osteolysis was induced by implantation of 6 mg fabricated wear of UHMWPE or VE-UHMWPE onto the surface of the calvarial bone of mice for 7 days[37]. Mice wild type and AnxA-1-deficient mice were anesthetized by intraperitoneal injection of 100 mg/kg ketamine and 10 mg/kg xylazine. Thereafter, sagittal incision (~1 cm) was made over the calvarial anterior site for implantation of fabricated wear UHMWPE or VE-UHMWPE particles. Depletion of neutrophils was performed in eight-week-old male BALB/c mice using 3 intraperitoneal injections of 150 μg anti-mouse Ly6G (Bio-X-Cell, NH, USA) on days −1, 2, and 5 post debris implantations[39]. Control mice received 3 intraperitoneal injections of 150 μg IgG2a isotype (Bio-X-Cell, NH, USA). This protocol was optimized to obtain a significant reduction in splenic Ly6G⁺ and CD11b⁺ (Biolegend) cells as analyzed flow cytometry[39]. Immunohistochemistry staining using an antibody to elastase (Abcam) was also used to confirm the depletion of neutrophils in the granulomatous tissue that was formed after the implantation of debris. For adoptive transfer experiment, bone marrow-derived neutrophils (BMN) were obtained from 8-week-old male C57/BL6 mice using mouse neutrophil isolation kit according to the

manufacturer's instruction (Miltenyi Biotec, Auburn, CA, USA). Cells purity was >97% neutrophils as analyzed by Giemsa-stained smears. Cell were cultured for 30 min in the presence or absence of 30 ng/mL mouse TNF-α (Biolegend), washed with ice-cold PBS and then administrated at $1 \times 10^6$ onto calvarial bone after the implantation of the UHMWPE debris. Cells without stimulation were considered as mock BMN and TNF-α-stimulated cells were considered as aBMN. Blocking AnxA1 in vivo was carried out in 8-week-old male C57BL/6 mice using 5 μg of a neutralizing mouse monoclonal antibody to AnxA1 (R&D System) that was subcutaneously injected onto calvarial bone at the time of debris implantation and on days 2 and 4 post-implantation. Following the same regime, control mice received 5 μg of an iso-type control antibody (R&D System). For examining the therapeutic effects of AnxA1, eight-week-old male C57/BL6 mice was treated by N-terminal AnxA1 (Ac2–26) or BML111 (R&D Systems) was injected 5 times at a concentration of 1 mg/kg. One group of mice were treated with 3 injections of 2.5 mg/kg WRW4 (R&D Systems) on days 1, 3, and 5 post debris implantations. Reagents were dissolved in water and volume and concentrations were adjusted in PBS to 100 μl. Injections were performed subcutaneously onto calvarial bones after implantation of debris. Sham mice and these received PBS injections were considered as controls. In separate experiment, eight-week-old male C57/BL6 mice were treated by 5 subcutaneous injection of Ac2–26 (Synthesized as described in Supplementary Fig. 10) and parallelly 5 intra-peritoneal injections of GW9662 (Cayman Chemical, Michigan, USA). GW9662 was dissolved in dimethyl sulfoxide (DMSO) and diluted with PBS to a final concentration of 10%. 1 mg/kg for GW9662 or its solvent (vehicle) was intraperitoneally injected every day for 5 days. For the cytokines-induced osteolysis model, murine recombinant RANKL or TNF-α (Biolegend, San Diego, USA) were injected onto the surface of the calvarial bone on 4 consecutive days at a concentration of 100 μg/kg. Pathological bone erosions were evaluated on day 7 day for debris-induced osteolysis and on day 5 for cytokine-induce models using high-resolution micro-computed tomography assessment (micro-CT), histopathology and gene expression.

**Micro-CT analysis and bone histomorphometry.** Micro-computed tomography assessment (micro-CT) of Calvariae was performed using R-mCT2 scan analysis (Rigaku, Tokyo, Japan). Images were analyzed using ImageJ (NIH, USA) for the quantification of bone loss on the surface of calvariae[37,38]. For histomorphometry, calvariae were then fixed in 10% of formalin for 48 h, decalcified in a Decalcifying Solution B (FUJIFILM Wako Pure Chemical Corporation, Osaka, Japan) at 4 °C for 3 days, and then embedded in paraffin for conducting the histological examination. Thereafter, 5 μm-sections were stained with leukocyte acid phosphatase tartrate resistance acid phosphatase (TRAP, Sigma, Tokyo, Japan) and hematoxylin eosin (HE, Wako). The lesions and infiltration of inflammation cells were microscopically examined, and images were analyzed with Image J (NIH) for quantitative evaluation[38].

**Western Blotting.** Proteins of lysed cells were separated in SDS-PAGE gels by electrophoresis and then transferred to polyvinylidene fluoride membrane (Immobilon-P Membrane; Merck, Darmstadt, Germany). Primary antibodies to β-actin (SP124, 1:2000 dilution, Abcam, UK), AnxA-1 (BL28553, 1:1000 dilution, Biolegend, San Diego, USA), total NFκB P65 (14G10A21, 1:1000 dilution, Biolegend, San Diego, USA), phospho NFκB P65 (GTX133899, 1:1000 dilution, Gene-Tex, CA, USA), phospho NFκB P105 (18E6, #4806, 1:1000 dilution, Cell signaling technology, CST, MA, USA), phospho P38 (D3F9, #4511, 1:1000 dilution, CST, MA, USA), phospho RelB (D41B9, #5025, 1:1000 dilution, CST, MA, USA) and PPAR-γ (D8I3Y, #95128, 1:1000 dilution, CST, MA, USA). Respective secondary antibodies, including anti-mouse HRP conjugated antibody (1:2000 dilution; CST, MA, USA) and anti-rabbit HRP conjugated antibody (1:2000 dilution; CST, MA, USA) were used for detection of bound antibodies. Signals were detected by Ez WestLumi Plus (ATTO, Tokyo, Japan) and bands were visualized using a Quantity One v. 4.6.9 (Bio-Rad) software. Bands were quantified for relative intensity using ImageJ software (NIH, USA).

**Osteoblast culture and stimulation.** Human fetal osteoblasts purchased from Cell Applications (San Diego, CA, USA) were cultured in osteoblast growth medium (Cell Applications). Cells were differentiated in osteoblast differentiation medium (Cell Applications) for 14 days and cultures were regularly replenished with fresh media every 3 days[40]. Cells were stimulated with either recombinant human TNF-α (Peprotech) for 48 h in the presence or absence recombinant AnxA1 (R&D Systems).

**Quantitative Real-Time Polymerase Chain Reaction (qRT–PCR).** Cells and tissues were lysed using TRIzol Reagent (Invitrogen) for RNA extraction and cDNAs synthesis using RNeasy Plus Mini kit columns (Qiagen, Hilden, Germany) and GoScript™ reverse transcriptase kit (Promega, Madison, USA), respectively. SYBR® Premix Ex Taq™ II (Takara, Shiga, Japan) was used for performing qRT-PCR with gene-specific primers described in Supplementary Table 2. Gene expression was calculated by the $2^{-\Delta\Delta Ct}$ method after normalizing to the GAPDH and β-actin[37].

**RNA sequencing and bioinformatics.** Human monocytes were cultured in a growth medium supplemented with recombinant macrophage colony-stimulating factor RANKL plus 100 ng/mL recombinant AnxA1 (R&D systems). We next

analyzed the transcriptional profiling of stimulated cells for 8 days by RNA sequencing. An average of 69 million reads (paired-end reads of 101 bp) per sample were mapped by the STAR software and read count was determined using the RSEM software. Significant differences were calculated using DESeq2 R package (https://www.r-project.org/). Analyses were performed using the Database for Annotation Visualization and Integrated Discovery online tools (DAVID: david.abcc.ncifcrf.gov). Heat map was used to visualize the differences in fold changes in each enriched GO term (http://biit.cs.ut.ee/clustvis/). The RNA-seq data are publicly available at the Gene Expression Omnibus (GEO) database (http: www.ncbi.nlm.nih.gov/geo/) under the accession numbers (GSE183145) and (GSE171542)[38].

**Preparation of thermo-responsive platform of the peptide.** Matrigel® (Corning Inc., USA) was thawed according to the manufacturer's guidelines and kept on ice during the experiments. A 750 µg sample of the lyophilized peptide was dissolved in 50 µL of ultra-pure distilled water and added drop-wise to the Matrigel® solution under vortexing. The resulting hydrogel was immediately either placed on ice or stored at 4 °C until used in further experiments. Aliquots of the prepared hydrogel were diluted in PBS (Wako, Japan) and assayed for the peptide content using a standard BCA assay method (Thermo Fisher scientific, USA). Peptide entrapment efficiency was calculated according to equation: Entrapment efficiency (%) = (Estimated peptide content/theoretical peptide content) × 100. For further evaluating the in vitro release of the hydrogel encapsulating the Ac2–26 peptide, the peptide-containing hydrogel was incubated at 37 °C for few minutes until solidification had occurred, and then incubated in PBS (pH 7.4) in a thermally controlled stirrer (37 °C, 300 rpm). Peptide release was quantified in samples (25 µL) that were frequently at specific time intervals. For a further understanding of the behavior of the prepared thermo-responsive hydrogel, a kinetic analysis of the release of the peptide from the hydrogel was applied. Linear regression analyses were utilized to fit the peptide release data to the most common models associated with the controlled-release dosage forms, including Zero-order (Equation: $M_t/M_\infty = K_0 t$.), First-order (Equation: $M_t/M_\infty = e^{-K1t}$) and Higuchi-diffusion (Equation: $M_t/M_\infty = k_H t^{1/2}$). The $M_t/M_\infty$ was the fractional release of the drug at time $t$, $k_0$ is the zero-order rate constant, $k_1$ is the first-order rate constant, $k_H$ is the Higuchi rate constant and $t$ is the time point at which release was estimated. In addition, the equation developed by Ritger and Peppas[41] was applied to elucidate the mechanism of peptide release from the hydrogel as Equation: $M_t/M_\infty = Kt^n$. The $M_t/M_\infty$ ratio was the fractional release of the drug at time $t$, $K$ was the release rate constant, and $n$ was the diffusional exponent. In the case of hydrogels formed as a thin film sample, $n = 0.5$ for Fickian diffusion; $0.5 < n < 1$ for non-fickian pattern; $n = 1$ for case II transport, and $n > 1$ for supercase II transport. Our results showed that formulation containing 600 µL Matrigel® achieved a sustain peptide release for one week, with a constant time-independent controlled-release pattern (Supplementary Table 3, 4). This behavior met our criteria and is highly favorable for ensuring a sustained therapeutic effect of a drug in vivo.

**Statistics and reproducibility.** Statistical analyses were performed using one-way ANOVA followed by Tukey's multiple comparisons procedure (multiple groups) and unpaired Student's $t$-test (two groups) to compare the differences among groups. The statistical tests used in each panel are mentioned in the figure legends. Results were considered statistically significant when $*p < 0.05$, $**p < 0.001$, $***p < 0.0001$, $****p < 0.00001$. ns indicates no significant difference. Error bars indicate SEM. Data were analyzed using GraphPad Software Prism 9 (CA, USA). Experiments were at least repeated three times for reproducibility of data. All quantitative experiments were evaluated by two researchers independently (blinded to experiment groups). The transmission electron microscope, micro-CT, and histological images are shown as representative images.

**Reporting summary.** Further information on research design is available in the Nature Research Reporting Summary linked to this article.

## Data availability
The bulk RNA-seq data in this paper were deposited on NCBI Gene Expression Omnibus (GEO) and are accessible through GEO Series accession numbers; GSE183145 and GSE171542. The remaining data are available within the article, the Supplementary Information or Source Data file provided with this paper. Source data are provided with this paper.

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

## Acknowledgements

This work was supported by research grants from Japan Society for the Promotion of Science (Grant-in-Aid for Scientific Research C; 17K10993), Akiyama Life Science Foundation, Kobayashi Foundation, Uehara Memorial Foundation, and the Japan Agency for Medical Research and Development (JP20gm6210004). We thank Dr. Keita Uetsuki and Mrs. Tomoyo Yutani from Teijin Nakashima Medical Co., Ltd. (Japan) for providing UHMWPE particulate debris from the hip implant materials. We also thank the technical assistance of Chowdhury Arpan in the synthesis and characterization of the peptide Ac2–26. Knockout mice (AnxA1 KO mice) were generated at the Laboratory Animal Resource Center, University of Tsukuba.

## Author contributions

Conceptualization: M.T., N.I., Methodology: H.A., M.A., T.S., G.M., Y.T., T.E., S.Y., Y.N., F.M., M.E., M.Y., H.H., Validation: M.T., H.A., K.K., D.T., Formal Analysis: H.A., M.A., G.M., T.E., Y.N., S.Y., Resources: M.T., T.S., D.T., Data Curation: M.T., H.A., T.S., Writing—Original Draft: H.A., Writing—Review & Editing: M.A., K.K., N.I., Visualization: M.T., H.A., G.M., Supervision: M.T., N.I., Project Administration: M.T., T.S., Funding Acquisition: M.T., T.S., K.K., D.T.

## Competing interests

The authors declare no competing interests.
