## [Peer Review File · Nature Communications]

Reviewers' comments:

Reviewer #1 (Remarks to the Author):

In this study Alhasan et al present a series of elegant experiments that identify neutrophils and neutrophil-derived Annexin A1 (AnxA1) as a major determinant in regulating immune cell reactivity in response to debris released from materials derived from prosthetic components. As such the main hypothesis behind the study is novel and it has been challenged in a very logical manner, applying multiple experimental approaches. All in all, the manuscript is very interesting and its take-home message very clear. Harnessing the biology discovered by the Authors with a material preparation of peptide Ac2-26 is another positive element of the study.

Specific Points (mainly minor)

1. The regulatory role of neutrophils in experimental disease settings is now emerging, and being reported in distinct studies. The ability of neutrophils to 'inform' and adjust the environment through their death by apoptosis, release of EVs or NET is also emerging, as a set of non-redundant phenomena. It would be complementary to the current dataset to explore and possibly report AnxA1 expression in human tissue as shown in Figure 1A and 1B.

2. The detection of EV which are AnxA1 positive (Figure 4D) is in-line with the original study of Dalli et al (PMID: 18594025) and the current identification of this protein as an efficient marker of EVs, mainly membrane-borne more than exosomes (PMID: 30951670). However the Authors have not followed up on these data: for instance, are neutrophil-derived EV taken up by the bone cells in their in vitro assays? If so with which kinetics. And this uptake, if it does happen, is specific to a certain degree, for neutrophil EVs as compared to EVs from other sources, for example platelet EVs, which do not contain AnxA1. In my view this information would add to the proposed mechanism by which neutrophil can regulate macrophages and bone cells. Also, would soluble or EV AnxA1 important here? This could be determined by removing EVs from neutrophil supernatants. Similarly, if platelet EVs are not taken up, can they be used as a negative control for the overarching hypothesis and model put forward with this study?

In short, some more details of the proposed biological circuit centred on neutrophils and AnxA1 would augment the impact and quality, which are already high, of the study.

3. There is no doubt that running the in-vivo model of osteolysis model will add to the strength of the message of this study. Even a global AnxA1 null mouse will be sufficient to reinforce the validity of the biological process identified here.

4. The Authors refer to recent studies linking AnxA1 to AMPK signalling in controlling macrophage skewing. Here they present strong evidence between AnxA1 and PPARgamma, with clear over-expression at 3 hours. In view of the characterised post-receptor signalling following AnxA1 interaction with FPR2, can they shed some light of what is upstream of PPARgamma induction? Such information will add to the current knowledge of the post-FPR2 signalling associated with specific pro-resolving properties of AnxA1.

5. Page 11. Line 233. Please remove the word 'systemic'.

6. Page 11, Line 238. It is wrongly reported that neutrophils can express IL-10. This is true for mouse neutrophils perhaps (paper of Cerundolo V in Nature Immunology; PMID: 20890286) but false for human neutrophils, where the IL-10 gene is packed so that it cannot be activated (Cassatella M in J Immunology; PMID: 23355741). Please correct this mistake or qualify it. Easier to remove IL-10 from the sentence.

7. Page 11, Line 236. Nadkarni et al. have reported in a previous publication than reference 14, the ability of neutrophils to promote a T helper phenotype (PMID: 27956610).

8. Page 14, Line 299. Of interest, RvD1 also activates FPR2. Have the Authors tested RvD1 or Lipoxin A4 in their in vitro assay for PPARgamma induction? This could be interesting and broaden the possibility to activate this protective circuit.

9. Page 14, Line 308. I am not sure references 29-32 are those supposed to be quoted here. Ref 29 refers to rosiglitazone, There is no reference for arthritis and peritonitis. Perhaps Dufton et al. (PMID: 20107188).

10. Figure 5. Indicate the concentrations of the agents used in Panels A-C. Is the rationale for the use of BML-111 explained and justified?

11. Is there a concentration-response curve with peptide Ac2-26 for instance in the PPARgamma experimental model in vitro, to have an idea of its potency? Side by side comparison with AnxA1 will be preferable.

Reviewer #2 (Remarks to the Author):

General comments:

This seems like a high-quality manuscript, with clear hypotheses, easy to follow trail of thought and clear writing/figures. The first part of the manuscript (data presented in the figures 1-3) aiming to show that neutrophils have a protective role in the wear particle induced osteolysis, is not clinically credible and considering the highly surprising nature of the findings not supported by enough experimental data. This section needs extensive revision and stronger experimental data.

The second part of the manuscript (data in the figures 4-8) showing that Anxa A1 has anti-osteoclast activity and that it protects against wear particle induced and inflammatory osteolysis, is both rationally and experimentally stronger. This part is mostly novel and of obvious interest to the field of bone biology.

Specific comments:

1) In contrast to the authors claims there are very few neutrophils present in the clinical aseptic loosening. This has been documented by numerous studies and, indeed, the increased amount of neutrophils is commonly used as an histopathological sign of a (subclinical) implant infection (Zmistowski B, et al. J Orthop Res. 2014 Jan;32 Suppl 1:S98-107). It seems very unlikely that the few neutrophils normally present in the aseptic loosening could play any significant role in the process.

2) The calvarial model is widely utilized to study wear particle induced osteolysis. However, I do feel that for this particular research question it's a poor choice. Surgical implantation of a large amount of UHMWPE particles undoubtedly induces an acute inflammatory reaction in which the neutrophils might participate in. In contrast, the clinical aseptic loosening is caused by slow accumulation of particles and chronic macrophage dominated inflammation that probably has nothing to do with neutrophils.

3) It would be crucial to know if the inflammatory infiltrate on the calvarium contained any neutrophils prior to depletion? In vivo imaging and/or immunostainings would need to be performed.

4) Similarly, successful depletion of the neutrophils at calvarium should be demonstrated by immunostainings.

5) Did the authors consider the possibility that Anxa A1 is released from macrophages? Performing immunostainings as well as additional in vitro experiments to identify its source would be helpful.

6) To comprehensively show that Anxa A1 plays a protective role in the aseptic loosening, experiments with Anxa A1 KO mouse are needed at this level.

6) UHMWPE particles are less dense than water and float in the cell culture experiments preventing direct contact with the cells. This is a well-recognized problem in the field and needs to be taken in to account when planning cell stimulation experiments. Surprisingly, it seems that this problem has not been addressed thus putting all the in vitro particle stimulation results in question. How can the particles have an effect on the cells if they float on the surface of the cell culture media?

7) How do the authors explain that Ac2-26 reduced osteolysis and other variables but BML111 had no effect?

8) The anti-osteoclast activity of Anxa A1 signaling pathway has been described once before and should be cited and appropriately discussed. Kao W et al. A formyl peptide receptor agonist suppresses inflammation and bone damage in arthritis. *Br J Pharmacol.* 2014;171:4087-96.

9) The method for generating UHMWPE particles needs to be described in more detail. A more detailed characterization of the particles would also be appreciated.

10) The neutrophil depletion method needs reference.

Reviewer #3 (Remarks to the Author):

Alhasan et al. examined the contribution of neutrophils to osteolysis. Osteolysis leads to failure of joint replacement in arthritic patients. Osteolysis is caused by inflammation. To test the role of neutrophils, Alhasan et al. used a particulate polyethylene debris-induced mouse model that produces osteolysis of calvaria. The authors show that depletion of neutrophils increased osteolysis suggesting that neutrophils

play a regulatory role. Using this model, the authors have identified that neutrophils stimulated by debris up-regulated AnxA1. Furthermore AnxA1 down regulated NF-kB signaling in TNF-induced and debris-induced activation of human macrophages. Finally, the authors that administration of a peptide mimetic derived from AnxA1 reduces inflammation in the calvaria debris-induced osteolysis model.

this is an interesting study. The results of this paper have potential clinical relevance, but there are issues that need to be addressed.

Major concerns:

1. The results of the histochemistry in (human) patients are not very convincing. The number of neutrophils in synovial fluid or synovium (Fig. 1) do not support an increased level of neutrophils. This makes the rest of the paper using the mouse model less interesting because the relevance is not clear. Furthermore, multiple cell types express GR1 marker (though MPO is highly expressed in neutrophils). (the quality of Fig 1A and 1B could be improved).
2. In Fig. 2 the authors use Ly6G antibody for depletion in the mouse model. This marker is also expressed on multiple cell types.
3. What is the mechanism by which AnxA1 mediates its function(s). The authors show an inhibitory effect of AnxA1 on NF-kB in pre-osteoclasts. Does AnxA1 work via a receptor or enter the cell directly? AnxA1 can also have an effect on inflammation, why only focus on the osteoclasts?

Minor Points:

1. The use of fluorescent TRAP substrate like ELF-97 may be helpful in quantifying data.
2. Can the authors justify why they used TNFa in experiment shown in Fig 8 and not debris model?

Reviewers' comments

We are grateful to the three reviewers for the constructive comments and suggestions that helped us to significantly improve our manuscript. We have performed all proposed experiments and revised the manuscript considering all comments and suggestions.

Reviewer #1 (Remarks to the Author):

In this study Alhasan et al present a series of elegant experiments that identify neutrophils and neutrophil-derived Annexin A1 (AnxA1) as a major determinant in regulating immune cell reactivity in response to debris released from materials derived from prosthetic components. As such the main hypothesis behind the study is novel and it has been challenged in a very logical manner, applying multiple experimental approaches. All in all, the manuscript is very interesting and its take-home message very clear. Harnessing the biology discovered by the Authors with a materiel preparation of peptide Ac2-26 is another positive element of the study.

Specific Points (mainly minor)

1. The regulatory role of neutrophils in experimental disease settings is now emerging, and being reported in distinct studies. The ability of neutrophils to 'inform' and adjust the environment through their death by apoptosis, release of EVs or NET is also emerging, as a set of non-redundant phenomena. It would be complementary to the current dataset to explore and possibly report AnxA1 expression in human tissue as shown in Figure 1A and 1B.

We acknowledge the reviewer comment and provide a new data showing stained sections with AnxA1. Please refer to Figure 4.

2. The detection of EV which are AnxA1 positive (Figure 4D) is in-line with the original study of Dalli et al (PMID: 18594025) and the current identification of this protein as an efficient marker of EVs, mainly membrane-borne more than exosomes (PMID: 30951670). However the Authors have not followed up on these data: for instance, are neutrophil-derived EV taken up by the bone cells in their in vitro assays? If so with which kinetics. And this uptake, if it does happen, is specific to a certain degree, for neutrophil EVs as compared to EVs from other sources, for example platelet EVs, which do not contain AnxA1. In my view this information would add to the proposed mechanism by which neutrophil can regulate macrophages and bone cells. Also, would soluble or EV AnxA1 important here? This could be determined by removing EVs from neutrophil supernatants. Similarly, if platelet EVs are not taken up, can they be used as a negative control for the overarching hypothesis and model put forward with this study? In short, some more details of the proposed biological circuit centred on neutrophils and AnxA1 would augment the impact and quality, which are already high, of the study.

We agree with the reviewer about the importance of addressing whether soluble or EV AnxA1 is important in our models. Therefore, we performed an in vivo study using function blocking antibody of AnxA1 in murine osteolysis model to address the reviewer comment. Our data showed that mice treated locally with AnxA1 antibody exhibited larger and greater osteolytic lesions than these in mice received control antibody. Results were shown in Figure 3. These results suggested that soluble AnxA1 is important for regulating osteolysis.

3. There is no doubt that running the in-vivo model of osteolysis model will add to the strength of the message of this study. Even a global AnxA1 null mouse will be sufficient to reinforce the validity of the biological process identified here.

We acknowledge the reviewer comment and we have generated knockout mice and performed the experiment in KO mice. Data are added to the Figure 3. Results and detailed methods were added to the manuscript.

4. The Authors refer to recent studies linking AnxA1 to AMPK signalling in controlling macrophage skewing. Here they present strong evidence between AnxA1 and PPARgamma, with clear over-expression at 3 hours. In view of the characterised post-receptor signalling following AnxA1 interaction with FPR2, can they shed some light of what is upstream of PPARgamma induction? Such information will add to the current knowledge of the post-FPR2 signalling associated with specific pro-resolving properties of AnxA1.

We acknowledge the reviewer comment and we have added the data of WRW4 treatment showing that this treatment blocking of FPR2 slightly reduced the increased expression of PPARG in human macrophages. Please refer to the supplementary information Figure 6.

5. Page 11. Line 233. Please remove the word 'systemic'.

We acknowledge the reviewer comment and we have deleted the word 'systemic' as requested.

6. Page 11, Line 238. It is wrongly reported that neutrophils can express IL-10. This is true for mouse neutrophils perhaps (paper of Cerundolo V in Nature Immunology; PMID: 20890286) but false for human neutrophils, where the IL-10 gene is packed so that it cannot be activated (Cassatella M in J Immunology; PMID: 23355741). Please correct this mistake or qualify it. Easier to remove IL-10 from the sentence.

We acknowledge the reviewer comment and we have deleted the IL-10 gene as requested.

7. Page 11, Line 236. Nadkarni et al. have reported in a previous publication than reference 14, the ability of neutrophils to promote a T helper phenotype (PMID: 27956610).

We acknowledge the reviewer comment and we have added the suggested reference as requested. Ref 14 Page 30.

8. Page 14, Line 299. Of interest, RvD1 also activates FPR2. Have the Authors tested RvD1 or Lipoxin A4 in their in vitro assay for PPARgamma induction? This could be interesting and broaden the possibility to activate this protective circuit.

That is very interesting, but this might be an extent of our study since the goal of our study is to investigate the protective role of AnxA1 in inflammatory osteolysis. Our data showed that AnxA1 is associated with activation of PPARG and reduction of NFkB signaling. In a recent related study, Xia et al., showed that RvD1 alleviates lung injury in mice by activating PPARG signaling pathway (10.1155/2019/6254587. eCollection 2019.).

9. Page 14, Line 308. I am not sure references 29-32 are those supposed to be quoted here. Ref 29 refers to rosiglitazone, There is no reference for arthritis and peritonitis. Perhaps Dufton et al. (PMID: 20107188).

We acknowledge the reviewer comment and we have modified the statement and added the suggested reference as requested. Ref. 34, Page 32.

10. Figure 5. Indicate the concentrations of the agents used in Panels A-C. Is the rationale for the use of BML-111 explained and justified?

We acknowledge the reviewer comment and we have added the concentration of each reagent to figure legend as requested. In addition, we indicated that BML-111 is agonist of FPR2. We used to compare the effect its resolving effect to Ac2-26, and we found that Ac2-26 is more effective. Please refer to figure legends Fig. 5, Page 36. In the discussion section, we indicated that AnxA1 and Ac2-26 in addition to their ability to activate FPR2, they might also interact with macrophages $\alpha 4\beta 1$ integrin and interfere in adhesion and migration based on the earlier study Ref. 32. Please refer to Page 15, Lines 330-337.

11. Is there a concentration-response curve with peptide Ac2-26 for instance in the PPARgamma experimental model in vitro, to have an idea of its potency? Side by side comparison with AnxA1 will be preferable.

We acknowledge the reviewer comment and we have added the data for stimulation with Ac2-26 to the supplementary materials. Results of AnxA1 stimulation are shown in the supplementary information Figure 8.

Reviewer #2 (Remarks to the Author):

General comments:

This seems like a high-quality manuscript, with clear hypotheses, easy to follow trail of thought and clear writing/figures. The first part of the manuscript (data presented in the figures 1-3) aiming to show that neutrophils have a protective role in the wear particle induced osteolysis, is not clinically credible and considering the highly surprising nature of the findings not supported by enough experimental data. This section needs extensive revision and stronger experimental data. The second part of the manuscript (data in the figures 4-8) showing that Anxa A1 has anti-osteoclast activity and that it protects against wear particle induced and inflammatory osteolysis, is both rationally and experimentally stronger. This part is mostly novel and of obvious interest to the field of bone biology.

Specific comments:

1) In contrast to the authors claims there are very few neutrophils present in the clinical aseptic loosening. This has been documented by numerous studies and, indeed, the increased amount of neutrophils is commonly used as an histopathological sign of a (subclinical) implant infection (Zmistowski B, et al. J Orthop Res. 2014 Jan;32 Suppl 1:S98-107). It seems very unlikely that the few neutrophils normally present in the aseptic loosening could play any significant role in the process.

We agree with the reviewer that few neutrophils may have minor role in disease control and regulation. However, we believe that the appearance of neutrophils in tissues is dependent on disease progression and stage. In addition, neutrophils have short life in tissues, which makes it is detection in tissue difficult in some cases. Neutrophils were detected in our model after inducing acute inflammation by implantation large amount of debris. Most probably, neutrophils infiltrate in the tissue at early stage to interact with wear debris, and their number gradually decreased in granulomatous tissue as the progress to chronic inflammation. Importantly, our model leads to identification of AnxA1 for treatment of inflammatory osteolysis. We further provided solid evidence on the therapeutic effects of AnxA1 Ac2-26 in inflammatory osteolysis and pathological bone resorption.

2) The calvarial model is widely utilized to study wear particle induced osteolysis. However, I do feel that for this particular research question it's a poor choice. Surgical implantation of a large amount of UHMWPE particles undoubtedly induces an acute inflammatory reaction in which the neutrophils might participate in. In contrast, the clinical aseptic loosening is caused by slow accumulation of particles and chronic macrophage dominated inflammation that probably has nothing to do with neutrophils.

We agree with reviewer that implantation of a large amount of UHMWPE particles induces an acute inflammatory reaction while clinical aseptic loosening is caused by slow accumulation

of particles and macrophages mediating chronic inflammation. However, current osteolysis model remains the most used model to evaluate therapeutic effects of agents in treatment and control. Our data showed that neutrophils are present in granulomatous tissue of patients who underwent revision surgery. In addition, depletion and adoptive transfer experiments showed strong evidence on the importance of the cells in the control of inflammatory osteolysis. We provided new images showing the presence of stained neutrophils in granulomatous tissues collected from clinical and experimental samples using antibody to MPO and neutrophil elastase. Please refer to Figure 1.

3) It would be crucial to know if the inflammatory infiltrate on the calvarium contained any neutrophils prior to depletion? In vivo imaging and/or immunostainings would need to be performed.

We acknowledge the reviewer comment and we have added figure showing the presence of neutrophils on the calvarium after implantation of wear debris. Please refer to the supplementary information Figure 10.

4) Similarly, successful depletion of the neutrophils at calvarium should be demonstrated by immunostainings.

We acknowledge the reviewer comment and we have added figure to the supplementary information for successful depletion of the neutrophils in calvarium as requested. Please refer to the supplementary Figure 10.

5) Did the authors consider the possibility that Anxa A1 is released from macrophages? Performing immunostainings as well as additional in vitro experiments to identify its source would be helpful.

We acknowledge the reviewer comment and we have added figure for AnxA1 staining using clinical samples. In addition, we performed Western blot analysis for macrophages, and we found that AnxA1 expression didn't significantly increased after stimulation with wear debris. Slight increase in AnxA1 expression is in M2 differentiated macrophages as compared to M0 and M1 macrophages. Please refer to the supplementary information Figure 3.

6) To comprehensively show that Anxa A1 plays a protective role in the aseptic loosening, experiments with Anxa A1 KO mouse are needed at this level.

We acknowledge the reviewer comment and we have generated knockout mice and performed the experiment in KO mice. Data are added to the Figure 3. Results and detailed methods were added to the manuscript.

6) *UHMWPE particles are less dense than water and float in the cell culture experiments preventing direct contact with the cells. This is a well-recognized problem in the field and needs to be taken in to account when planning cell stimulation experiments. Surprisingly, it seems that this problem has not been addressed thus putting all the in vitro particle stimulation results in question. How can the particles have an effect on the cells if they float on the surface of the cell culture media?*

We agree with reviewer that traditional culture experiment is not suitable for stimulation of macrophages. Thus, in our experiment for stimulating macrophage with UHMWPE particles, we used inverted method that allow particles to directly interact with macrophages. We clearly indicated that we used inverted culture model in the Methods section. Please refer to Page 24, Lines 515-516.

7) *How do the authors explain that Ac2-26 reduced osteolysis and other variables but BML111 had no effect?*

This is very important comment. It is known that BML111 activate FPR2 signaling, however, Ac2-26, in addition to its ability to activate FPR2, it inhibits cell adhesiveness and migration via downmodulating $\alpha 4\beta 1$ integrin and their affinity and valency, without changing their cell surface expression. Therefore, it is possible that the therapeutic effects of Ac2-26 in inflammatory osteolysis might be due to its ability to reduce integrin-dependent monocyte adhesion and the migration necessary for the development of inflammation and osteoclast formation. We indicated this information in the Discussion section. Page 15, Lines 335-337.

8) *The anti-osteoclast activity of Anxa A1 signaling pathway has been described once before and should be cited and appropriately discussed. Kao W et al. A formyl peptide receptor agonist suppresses inflammation and bone damage in arthritis. Br J Pharmacol. 2014;171:4087-96.*

We acknowledge the reviewer comment and we have added this information and cited the article accordingly. Ref. 15, Page 30.

9) *The method for generating UHMWPE particles needs to be described in more detail. A more detailed characterization of the particles would also be appreciated.*

We acknowledge the reviewer comment and we have added more detailed characterization of particles as requested. Page 18, Lines 386-399.

10) *The neutrophil depletion method needs reference.*

We acknowledge the reviewer comment and we have added a reference. Ref. 41, Page 33.

Reviewer #3 (Remarks to the Author):

Alhasan et al. examined the contribution of neutrophils to osteolysis. Osteolysis leads to failure of joint replacement in arthritic patients. Osteolysis is caused by inflammation. To test the role of neutrophils, Alhasan et al. used a particulate polyethylene debris-induced mouse model that produces osteolysis of calvaria. The authors show that depletion of neutrophils increased osteolysis suggesting that neutrophils play a regulatory role. Using this model, the authors have identified that neutrophils stimulated by debris up-regulated AnxA1. Furthermore AnxA1 down regulated NF- κ B signaling in TNF-induced and debris- induced activation of human macrophages. Finally, the authors that administration of a peptide mimetic derived from AnxA1 reduces inflammation in the calvaria debris-induced osteolysis model. This is an interesting study. The results of this paper have potential clinical relevance, but there are issues that need to be addressed.

Major concerns:

1. The results of the histochemistry in (human) patients are not very convincing. The number of neutrophils in synovial fluid or synovium (Fig. 1) do not support an increased level of neutrophils. This makes the rest of the paper using the mouse model less interesting because the relevance is not clear. Furthermore, multiple cell types express GR1 marker (though MPO is highly expressed in neutrophils). (the quality of Fig 1A and 1B could be improved).

We acknowledge the reviewer comment and we have added new images for the stained neutrophils with specific markers as requested. Please refer to the Figure1.

2. In Fig. 2 the authors use Ly6G antibody for depletion in the mouse model. This marker is also expressed on multiple cell types.

Ly6G-specific mAb, clone 1A8 has been broadly used to deplete neutrophils unlike clone RB6-8C5 that also recognize dendritic cells, and subpopulations of lymphocytes and monocytes (<https://bxcell.com/product/invivoplus-anti-m-ly-6g-2/>). Here are some studies in which authors used the antibody:

Daley et al., *J Leukoc Biol.* 2008;83(1):64-70. doi: 10.1189/jlb.0407247. PMID: 17884993 & Moynihan et al., *Nat Med* 2016; 1402–1410. <https://doi.org/10.1038/nm.4200>

3. What is the mechanism by which AnxA1 mediates its function(s). The authors show an inhibitory effect of AnxA1 on NF- κ B in pre-osteoclasts. Does AnxA1 work via a receptor or enter the cell directly? AnxA1 can also have an effect on inflammation, why only focus on the osteoclasts?

This is important point of our study. We focused on osteoclast because they are the main players in pathological bone resorption. We have showed that AnxA1 reduced NF- κ B in

macrophages that associated with reduction of inflammation and osteoclast differentiation. This is due to its ability to activate PPARG pathway. We have added data showing that inhibiting of FPR2 partly reduced the expression of PPARG in macrophages stimulated with AnxA1, which may suggest the importance of AnxA1/FPR2/PPARG axis in inhibiting inflammation and osteoclast differentiation.

Minor Points:

1. The use of fluorescent TRAP substrate like ELF-97 may be helpful in quantifying data.

We agree with reviewer about the usefulness of fluorescent TRAP substrate like ELF-97 for quantifying osteoclasts. However, the method we have utilized for quantifying osteoclasts is extensively used in the osteoclast research area.

2. Can the authors justify why they used TNFa in experiment shown in Fig 8 and not debris model?

Thank you very much for your comment. In fact, we used TNFa model because of the following reasons:

1. It is evident that macrophages produce TNF-a in response to stimulation with implant debris and TNF-a is the major cytokines associated with promotion of inflammatory osteolysis associated with implant loosening. In fact, TNF-a and RANKL are major cytokines responsible for the development of inflammatory osteolysis and aseptic loosening.
2. Our data showed that AnxA1 treatment inhibited inflammation induced by TNF-a and osteoclast differentiation mediated by RANKL in vitro and we wanted to examine the effect in vivo to gain an insight into the molecular function.
3. Our developed hydrogel is solidified in vivo (at 37°C) and the debris might be captured by the gel if we use debris induced osteolysis model, and this may influence the interaction between debris and immune cells.

REVIEWERS' COMMENTS

Reviewer #2 (Remarks to the Author):

Authors have done a good job revising the manuscript and have sufficiently addressed all my concerns. I have no further comments.

Response to Reviewers

Reviewer #1 (Remarks to the Author):

The Authors have addressed my concerns or comments satisfactorily. In fact, I appreciate them going the extra-mile and generating new tools to answers my scientific queries. I have no further points to raise.

We thank the reviewer for this assessment and the useful and constructive suggestions made during the first round of review.

Reviewer #2 (Remarks to the Author):

Authors have addressed some of the reviewers' comments and the manuscript has somewhat been improved by the revision. As was the case in the original submission authors do a good job of showing the modulatory role of AnxA1 in the wear particle induced osteolysis and demonstrate its therapeutic potential (Data presented in figures 3F to figure 8). I have no problem with this part of the manuscript and feel that its relevant and interesting. Unfortunately, the main issue of the original manuscript also persists, i.e. the claim that neutrophils play a modulatory role in the wear particle induced osteolysis is simply not credible and not supported by sufficient data. Another major issue is the source of AnxA1 that has not been sufficiently demonstrated by the current experiments.

We acknowledge the reviewer comments and we have focused our manuscript on the main findings demonstrating the modulatory role of AnxA1 in the wear particle induced osteolysis and its therapeutic potential. In addition, we provided data showing the cells expressing AnxA1 in periprosthetic tissues.

1) The fact that neutrophil count is clinically used marker for implant related infection (Parvizi et al 2011) and that aseptic loosening typically contains very few neutrophils needs to be thoroughly addressed and discussed. How can the small number of neutrophils present in the clinical aseptic loosening have any significant role in the pathogenesis of the loosening? Authors reply to this major question is vague and certainly not sufficient nor is their analysis of three patient samples. Indeed, there's a wealth of prior histological studies demonstrating the lack of neutrophils in the aseptic loosening and analyzing three patient samples does not change this fact. The very least authors need to acknowledge this fundamental problem in their reasoning, explain how they expect the in vivo results to be generalizable to the clinical condition, and revise the manuscript accordingly. Parvizi J, et al. New definition for periprosthetic joint infection: from the Workgroup of the Musculoskeletal Infection Society. Clin Orthop Relat Res. 2011;469:2992-4.

We totally agree that number of neutrophils in periprosthetic tissues is low compared to macrophages. However, we could detect significant number of elastase positive cells in the tissues, and we confirmed that the three cases included in this study have no infection. Considering the important point raised by the reviewer, we have focused our manuscript on the

main findings demonstrating the modulatory role of AnxA1 in the wear particle induced osteolysis and its therapeutic potential. The manuscript was reorganized accordingly.

The objective of the study in the introduction was changed as follows: “Nonetheless, there is a growing body of evidence to suggest that controlling chronic inflammation at the site of an implant would be a promising approach for therapeutic intervention⁷. Chronic inflammation generally occurs when the initial acute inflammation is not effectively resolved due to the inadequate pro-resolving activity of immune system, a process that is referred to as frustrated resolution. The resolution of inflammation is an active process that is rigidly orchestrated by endogenous pro-resolving mediators that function not as immunosuppressive agents, but instead they promote the resolution of inflammation through activating homeostatic control mechanisms in the affected tissues⁸⁻¹⁰. Of these molecules, Annexin A1 (AnxA1), a member of the annexin superfamily that is mainly released by monocytes and neutrophils, has been implicated in number of biological processes, including inflammation, intracellular vesicle trafficking, leukocyte migration, and tissue growth and regeneration, and apoptosis¹¹. In fact, the ability of AnxA1 to stimulate endogenous pro-resolving pathways leading to tissue repair and healing and its therapeutic effects have been documented in a broad range of experimental models, including myocardial ischemia injury, stroke, sepsis, arthritis, and multiple sclerosis¹².”

Given that periprosthetic osteolysis is a chronic inflammatory disorder typified by persistent inflammation, and that pro-resolving mediators may restore tissue homeostasis, we explored the function of AnxA1 in the pathophysiology of disease and evaluated its therapeutic applications in experimental periprosthetic osteolysis models.” Page 4, Lines 71-89.

2) Figure 1a, shows a large number of neutrophils in the patient samples. How was the implant related infection ruled out?

Thank you very much for pointing this out. We have discussed with our senior surgeons, and we confirmed that all samples have no infection. All three cases have more than 10 years after primary arthroplasty. We also checked their clinical records and data asked, and all cases showed low level of CRP. We indeed observed the presence of neutrophils in periprosthetic tissues in 3 different patients’ samples. However, based on the reviewer comments, we have focused our manuscript on the role of AnxA1 in the wear particle induced osteolysis and its therapeutic potential. Depletion and adoptive experiments were directed as a strategy to manipulate the local expression of AnxA1 in particle induced osteolysis models. We have modified the text and deleted the statement mentioning the potential role of neutrophils in the disease as recommended by the reviewer. In addition, we mentioned in the Method section that all cases have no infection and had low level of CRP.

3) Figure 1b, flow cytometry is the standard practice of analyzing the leukocyte populations in the synovial fluid. Why wasn’t this performed? Getting the actual count of neutrophils present in the synovial fluid would be more demonstrative of their potential significance.

We totally agree with reviewer that flow cytometry is the standard practice of analyzing the leukocyte populations in the synovial fluid. However, we usually can get very few amount of fluid from hip joint and pseudocapsule tissues around implant, around 200-500 ul, which contains very few number of cells not enough for performing flow cytometry. Therefore, we performed IFA for the isolated cells to confirm the presence of neutrophils. Considering the reviewer comment, we removed the data of IFA as we directed our manuscript to explore the role of AnxA1 in particle induced osteolysis.

4) It's not clear from the methods section if the inverted cell culture method used for all in vitro particle stimulation studies and not just for the macrophage stimulations?

We apologize for the incomplete information about the methods. All experiments for stimulating the cells, including macrophages, neutrophils and FLS with debris were performed using inverted method. We clearly described the method in the Methods section.

In the Method section for macrophages: "Differentiated macrophages were stimulated with UHMWPE debris in presence or absence of 100 µg recombinant human AnxA1 (R&D Systems, MN, USA) using an inverted culture system³." Page 19, Lines 416-417.

For neutrophils: "Freshly isolated neutrophils 1×10^6 were seeded onto poly-d-lysine-coated wells and stimulated with UHMWPE debris at a density of 0.1 mg/cm³ in minimum essential medium Eagle (MEM, Sigma) supplemented with 10% heat-inactivated fetal bovine serum (FBS, Nichirei Biosciences INC, Tokyo, Japan), and 5% mg/L penicillin/streptomycin solution (Wako, Japan) for 2 h at 37 °C in a humidified atmosphere containing 5% CO₂ using inverted culture system³." Page 18, Lines 389-390.

For hFLS: "The hFLS from normal healthy human synovial tissues purchased from Cell Applications (Cell Applications) were cultured according to the supplier's recommendations. Cultured cells were stimulated with UHMWPE debris at a density of 0.1 mg/cm³ for 24 h at 37 °C in a humidified atmosphere containing 5% CO₂ using inverted culture system³." Page 20, Line 440-443.

5) If inverted cell culture was used, I would expect to see phagocytosis of particles by neutrophils (figure 1D, E). Please comment.

We performed inverted method and we observed phagocytosis. We acknowledge the reviewer comments and we have added an image of phagocytosis of particles to Figure 1. Please see below images showing phagocytosis process of particles.

6) What tissue is shown in the figure 1C? It looks different from the other calvarian model pictures, for example figure 2D.

Images are prepared from same tissues, but magnification is different. In figure 1C, we want to show the infiltrated cells onto calvarial bone while in Figure 2D, we showed the pathological bone resorption, bone erosions and infiltrated cells onto calvarial bone. However, we provided better images with control sham as recommended by the reviewers. Please see figure 2.

7) Images from negative control mice (sham treated) should be shown for comparison the figure 1C.

We acknowledge the reviewer comments and we have added an image of sham as control. Usually, we don't observe any infiltrated cells in sham group. Please see figure 2.

8) Please show quantification of the neutrophils present in the histological sections from negative controls (sham), particle stimulation and neutrophil depletion model.

We acknowledge the reviewer comment and we have added quantification for cell count as

recommended by reviewer. Results are shown in Figure 2. Figure is shown below for review.

9) I feel successful neutrophil depletion (data now presented on the supplementary 10) should be shown in figure 2 as successful depletion is a critical for the interpretation of the results.

We acknowledge the reviewer comment and we have added it to Figure 2.

10) There's a number of reasons why neutrophil depletion (and adoptive transfer) could potentially impact the wear particle induced osteolysis in this mouse model. For example, did the authors consider the possibility that simply neutrophil apoptosis could be the factor downregulating the inflammation in their depletion and adoptive transfer models? Importantly, the current results only show association between neutrophils and AnxA1 production but do not demonstrate causality, i.e. that the particle activated neutrophils are the source of AnxA1.

We acknowledge the reviewer comment and we have modified the text as the current results only show association between neutrophils and AnxA1. In addition, we performed immunostaining with neutrophil and macrophage markers in clinical and experimental mouse model samples to understand the source of AnxA1.

11) In relation to comment #10, performing AnxA1 immunostainings, preferably double immunostaining with neutrophil and macrophage markers, from the in vivo model would be critical to demonstrate the cellular origin of the AnxA1.

We acknowledge the reviewer comment and we have performed immunostaining with neutrophil and macrophage markers in clinical and experimental mouse model samples to understand the source of AnxA1. Results are added to Figure 1 and Supplementary figure 1. Images were shown below for review.

Clinical samples

Experimental samples from debris-induced osteolysis

12) Please also provide more detailed analysis of the localization of AnxA1 in the clinical histological samples (Figure 3E), to what cell types is the AnxA1 staining localized to? Comparison to some other relevant tissue (e.g. synovium from osteoarthritis) would be helpful.

We acknowledge the reviewer comment and we performed immunostaining for OA hip samples before primary arthroplasty. Results are added to Supplementary figure 1. We found that AnxA1 is highly expressed in the lining cells. Images were shown below for review.

13) The western blot analysis demonstrated in the supplementary figure 3 would seem to demonstrate that macrophages produce AnxA1 even without any stimulation. This certainly does not rule out macrophages as a potential source of AnxA1, on the contrary. Please comment.

Thank you very much for the comment. It is well known that AnxA1 is produced mainly by innate immune cells including macrophages and neutrophils and the expression is increased upon specific stimulation. Our in vitro and staining data showed that both macrophages and neutrophils in tissues are expressing AnxA1. Therefore, we modified the text based on the current new data and concluded that macrophages and neutrophils are potential source of AnxA1.

14) Results, paragraph “Neutrophil-derived AnxA1 is a potential regulator of inflammation and pathological boneresorption induced by wear debris” could benefit from language revision. This is the first time AnxA1 is mentioned, and the reader is left wondering how did the authors decide to study this particular cytokine?

We acknowledge the reviewer comment and we have modified the statement as recommended. The new sentence is as follows: “AnxA1 is a potential inhibitor of inflammation and pathological bone resorption induced by wear debris.” Page 6, Lines 129-130.

15) The synovial fluid analysis (Figure 3D right panel) is of little use without comparison to some other synovial fluid sample. Do the authors expect that the AnxA1 production is increased or decreased in the aseptic synovial fluid e.g. compared to normal synovial fluid?

We agree that adding normal synovial fluid negative control is useful to compare the expression of AnxA1 in healthy and disease condition. However, it is difficult to obtain synovial fluid from healthy hip due to ethical and technical issues. Considering the reviewer comment, we added a new Figure to the Supplementary Figure 1 showing AnxA1 expression in synovial fluids of OA and aseptic loosening. We expect that expression of AnxA1 is increased in aseptic loosening in compassion to normal condition due to the elevation in inflammation. The increase of AnxA1 expression in damaged and inflamed tissues occurs as an attempt of the body to

restore the homeostatic control mechanisms in the affected tissues. Images for AnxA1 expression in synovial fluids of OA and aseptic loosening is shown below for review.

16) The in vivo experiments with WRW4 and BML11 (data presented in figure 5) would seem to suggest that the therapeutic effects of Ac2-26 are not mediated by the FPR2 receptor. Authors have added some discussion on the possibility that Ac2-26 might exert its effects via regulation of integrins but don't mention the WRW4 or BML11 results when discussing this topic. Please comment and revise accordingly.

We acknowledge the reviewer comment, and we have a new sentence to the Discussion section as follows: "In line with this supposition, neither the activation of FPR2 by an agonist treatment (BML11) nor blocking by an antagonist treatment (WRW4) had a significant impact on the pathological bone lesions induced by debris implantation, suggesting that the therapeutic potentials of Ac2-26 is independent form FPR2 signaling." Page 15, Lines 324-327.

Reviewer #3 (Remarks to the Author):

My main concerns were addressed by the authors in the revised manuscript. I do not have any further critiques.

We thank the reviewer for this assessment and the useful and constructive suggestions made during the first round of review.

Reviewers' comments:

Reviewer #1 (Remarks to the Author):

The Authors have addressed my concerns or comments satisfactorily. In fact, I appreciate them going the extra-mile and generating new tools to answers my scientific queries.

I have no further points to raise.

Reviewer #2 (Remarks to the Author):

Authors have addressed some of the reviewers' comments and the manuscript has somewhat been improved by the revision. As was the case in the original submission authors do a good job of showing the modulatory role of AnxA1 in the wear particle induced osteolysis and demonstrate its therapeutic potential (Data presented in figures 3F to figure 8). I have no problem with this part of the manuscript and feel that its relevant and interesting.

Unfortunately, the main issue of the original manuscript also persists, i.e. the claim that neutrophils play a modulatory role in the wear particle induced osteolysis is simply not credible and not supported by sufficient data. Another major issue is the source of AnxA1 that has not been sufficiently demonstrated by the current experiments.

1) The fact that neutrophil count is clinically used marker for implant related infection (Parvizi et al 2011) and that aseptic loosening typically contains very few neutrophils needs to be thoroughly addressed and discussed. How can the small number of neutrophils present in the clinical aseptic loosening have any significant role in the pathogenesis of the loosening? Authors reply to this major question is vague and certainly not sufficient nor is their analysis of three patient samples. Indeed, there's a wealth of prior histological studies demonstrating the lack of neutrophils in the aseptic loosening and analyzing three patient samples does not change this fact. The very least authors need to acknowledge this fundamental problem in their reasoning, explain how they expect the in vivo results to be generalizable to the clinical condition, and revise the manuscript accordingly.

Parvizi J, et al. New definition for periprosthetic joint infection: from the Workgroup of the Musculoskeletal Infection Society. Clin Orthop Relat Res. 2011;469:2992-4.

2) Figure 1a, shows a large number of neutrophils in the patient samples. How was the implant related infection ruled out?

3) Figure 1b, flow cytometry is the standard practice of analyzing the leukocyte populations in the synovial fluid. Why wasn't this performed? Getting the actual count of neutrophils present in the synovial fluid would be more demonstrative of their potential significance.

4) It's not clear from the methods section if the inverted cell culture method used for all in vitro particle stimulation studies and not just for the macrophage stimulations?

- 5) If inverted cell culture was used, I would expect to see phagocytosis of particles by neutrophils (figure 1D, E). Please comment.
- 6) What tissue is shown in the figure 1C? It looks different from the other calvarian model pictures, for example figure 2D.
- 7) Images from negative control mice (sham treated) should be shown for comparison the figure 1C
- 8) Please show quantification of the neutrophils present in the histological sections from negative controls (sham), particle stimulation and neutrophil depletion model.
- 9) I feel successful neutrophil depletion (data now presented on the supplementary 10) should be shown in figure 2 as successful depletion is a critical for the interpretation of the results
- 10) There's a number of reasons why neutrophil depletion (and adoptive transfer) could potentially impact the wear particle induced osteolysis in this mouse model. For example, did the authors consider the possibility that simply neutrophil apoptosis could be the factor downregulating the inflammation in their depletion and adoptive transfer models? Importantly, the current results only show association between neutrophils and AnxA1 production but do not demonstrate causality, i.e. that the particle activated neutrophils are the source of AnxA1.
- 11) In relation to comment #10, performing AnxA1 immunostainings, preferably double immunostaining with neutrophil and macrophage markers, from the in vivo model would be critical to demonstrate the cellular origin of the AnxA1.
- 12) Please also provide more detailed analysis of the localization of AnxA1 in the clinical histological samples (Figure 3E), to what cell types is the AnxA1 staining localized to? Comparison to some other relevant tissue (e.g. synovium from osteoarthritis) would be helpful.
- 13) The western blot analysis demonstrated in the supplementary figure 3 would seem to demonstrate that macrophages produce AnxA1 even without any stimulation. This certainly does not rule out macrophages as a potential source of AnxA1, on the contrary. Please comment.
- 14) Results, paragraph "Neutrophil-derived AnxA1 is a potential regulator of inflammation and pathological boneresorption induced by wear debris" could benefit from language revision. This is the first time AnxA1 is mentioned, and the reader is left wondering how did the authors decide to study this particular cytokine?
- 15) The synovial fluid analysis (Figure 3D right panel) is of little use without comparison to some other synovial fluid sample. Do the authors expect that the AnxA1 production is increased or decreased in the aseptic synovial fluid e.g. compared to normal synovial fluid?
- 16) The in vivo experiments with WRW4 and BML11 (data presented in figure 5) would seem to suggest that the therapeutic effects of Ac2-26 are not mediated by the FPR2 receptor. Authors have added some

discussion on the possibility that Ac2-26 might exert its effects via regulation of integrins but don't mention the WRW4 or BML11 results when discussing this topic. Please comment and revise accordingly.

Reviewer #3 (Remarks to the Author):

My main concerns were addressed by the authors in the revised manuscript. I do not have any further critiques.